# Impact of NO$_x$ on secondary organic aerosol (SOA) formation from α-pinene and β-pinene photo-oxidation: the role of highly oxygenated organic nitrates

Iida Pullinen[1,4], Sebastian Schmitt[1,5], Sungah Kang[1], Mehrnaz Sarrafzadeh[1,3,6], Patrick Schlag[1,7], Stefanie Andres[1], Einhard Kleist[2], Thomas F. Mentel[1], Franz Rohrer[1], Monika Springer[1], Ralf Tillmann[1], Jürgen Wildt[1,2], Cheng Wu[1,8], Defeng Zhao[1,9], Andreas Wahner[1] and Astrid Kiendler-Scharr[1]

[1]Institute for Energy and Climate Research, IEK-8, Forschungszentrum Jülich, 52425, Jülich, Germany
[2]Institute of Bio- and Geosciences, IBG-2, Forschungszentrum Jülich, 52425, Jülich, Germany
[3]Centre for Atmospheric Chemistry, York University, 4700 Keele St., Toronto, ON M3J 1P3, Canada
[4]Present address: Department of Applied Physics, University of Eastern Finland, Kuopio, Finland
[5]Present address: TSI GmbH, 52068 Aachen, Germany
[6]Present address: PerkinElmer, 501 Rowntree Dairy Rd, Woodbridge, ON, L4L 8H1, Canada
[7]Present address: Shimadzu Deutschland GmbH, 47269 Duisburg, Germany
[8]Present address: Department of Environmental Science, Stockholm University, 11418 Stockholm, Sweden
[9]Present address: Dept. of Atmos. and Oceanic Sci. & Inst. of Atmos. Sci., Fudan University, Shanghai, 200438, China

*Correspondence to*: Thomas F. Mentel (t.mentel@fz-juelich.de)

**Abstract.** The formation of organic nitrates (ON) in the gas phase and their impact on mass formation of Secondary Organic Aerosol (SOA) was investigated in a laboratory study for *α*-pinene and *β*-pinene photo-oxidation. Focus was the elucidation of those mechanisms that cause the often observed suppression of SOA mass formation by NO$_x$, and therein the role of highly oxygenated multifunctional molecules (HOM). We observed that with increasing NO$_X$ concentration: a) the portion of HOM organic nitrates (HOM-ON) increased, b) the fraction of accretion products (HOM-ACC) decreased and c) HOM-ACC contained on average smaller carbon numbers.

Specifically, we investigated HOM organic nitrates (HOM-ON), arising from the termination reactions of HOM peroxy radicals with NO$_x$, and HOM permutation products (HOM-PP), such as ketones, alcohols or hydroperoxides, formed by other termination reactions. Effective uptake coefficients $\gamma_{eff}$ of HOM on particles were determined. HOM with more than 6 O-atoms efficiently condensed on particles ($\gamma_{eff} > 0.5$ on average) and for HOM containing more than 8 O-atoms, every collision led to loss. There was no systematic difference in $\gamma_{eff}$ for HOM-ON and HOM-PP arising from the same HOM peroxy radicals. This similarity is attributed to the multifunctional character of the HOM: as functional groups in HOM arising from the same precursor HOM peroxy radical are identical, vapor pressures should not strongly depend on the character of the final termination group. As a consequence, the suppressing effect of NO$_X$ on SOA formation cannot be simply explained by replacement of terminal functional groups by organic nitrate groups.

According to their $\gamma_{eff}$ all HOM-ON with more than 6 O-atoms will contribute to organic bound nitrate (OrgNO$_3$) in the particulate phase. However, the fraction of OrgNO$_3$ stored in condensable HOM with molecular masses >230 Da appeared to be substantially higher than the fraction of particulate OrgNO$_3$ observed by aerosol mass spectrometry. This result suggests

losses of OrgNO$_3$ for organic nitrates in particles, probably due to hydrolysis of OrgNO$_3$ that releases HNO$_3$ into the gas phase but leaves behind the organic rest in the particulate phase. However, the loss of HNO$_3$ alone, could not explain the observed suppressing effect of NO$_X$ on particle mass formation from α-pinene and β-pinene.

Instead we can attribute most of the reduction in SOA mass yields with increasing NO$_X$ to the significant suppression of gas-phase HOM-ACC which have high molecular mass and are potentially important for SOA mass formation at low NO$_X$ conditions.

## 1 Introduction

Secondary organic aerosol (SOA) constitutes a substantial fraction of ambient aerosol. It is formed from oxidation products of volatile organic compounds (VOC) and known to adversely affect visibility, climate and human health (Hallquist et al., 2009). With annual emissions around 1100 Tg, the biosphere is the strongest source of tropospheric VOC (Guenther et al., 2012) and thus, SOA formation from biogenic VOC is of high importance. Despite the outstanding role of biogenic VOC by amount and reactivity, anthropogenic trace gases affect SOA formation and possible anthropogenic enhancement effects were found in laboratory and field studies (e.g., Carlton et al., 2010; De Gouw et al., 2005; Emanuelsson et al., 2013; Glasius et al., 2011; Hoyle et al., 2011; Shilling et al., 2013; Spracklen et al., 2011; Worton et al., 2011; Xu et al., 2015a). Examples of important anthropogenic trace gases are NO and NO$_2$, which together form the NO$_X$ family. During nighttime NO$_X$ is converted to NO$_3$ radicals, which oxidize biogenic VOC, leading to organic nitrates and SOA formation (Boyd et al., 2017; Boyd et al., 2015; Claflin and Ziemann, 2018; Faxon et al., 2018; Fry et al., 2013, 2014; Kiendler-Scharr et al., 2016; Lee et al., 2016a; Ng et al., 2017). During daytime, NO$_X$ controls the atmospheric HO$_X$ cycle thus the oxidation cycle of VOC by reaction with peroxy radicals. In this study we will focus on role of NO$_X$ for SOA formation during daytime. In a number of studies the role of NO$_X$ in the formation of SOA mass was investigated (Eddingsaas et al., 2012; Han et al., 2016; Kim et al., 2012; Kroll et al., 2006; Lee et al., 2016a; Lee et al., 2020; Ng et al., 2007; Pandis et al., 1991; Presto et al., 2005; Rindelaub et al., 2015, 2016; Sarrafzadeh et al., 2016; Stirnweis et al., 2017; Zhang et al., 2006 ). In most cases it was observed that NO$_X$ decreased mass yields of SOA formation and the effects were generally attributed to impacts of RO$_2\cdot$ + NO reactions. Sarrafzadeh et al. (2016) show that parts of the apparent suppression of SOA yields from β-pinene by NO$_X$ was due to the role of NO$_X$ in the HO$_X$ cycle. As mass yields of SOA formation from α-pinene and β-pinene photo-oxidation depend on the actual OH concentrations, NO$_X$ affects SOA formation also via decreasing or increasing OH concentrations according to reactions R1 and R2:

$$OH\cdot + NO_2 + M \quad \rightarrow \quad HNO_3 + M \qquad\qquad (R1)$$
$$HO_2\cdot + NO \quad \rightarrow \quad OH\cdot + NO_2 \qquad\qquad (R2)$$

NO$_X$ is inhibiting new particle formation (NPF, Wildt et al., 2014), therefore, in absence of seed particles, NO$_X$ can prevent formation of sufficient particle surface where low volatile compounds could condense on. In absence of particles other sinks gain in importance for low volatile compounds as dry deposition in the environment or wall losses in chamber experiments. In order to circumvent these effects, Sarrafzadeh et al. (2016) used seed particles always providing sufficient surface for the gas phase precursors of SOA mass to condense on and kept the OH concentrations constant. As a result the remaining effect of NO$_X$ on SOA mass formation from β-pinene and on SOA yields was only moderate. Generally, in absence of NO$_X$, peroxy radicals (RO$_2$·) mainly react with other peroxy radicals (including the HO$_2$· radical) whereby termination products like hydroperoxides, ketones, alcohols, carboxlic acids and percarboxylic acids are produced. In reactions between peroxy radicals also alkoxy radicals are formed, which continue the radical chain (R4b):

$$RO_2\cdot + HO_2\cdot \qquad \rightarrow \qquad ROOH + O_2 \qquad\qquad (R3a)$$

$$RC(=O)OO_2\cdot + HO_2 \qquad \rightarrow \qquad RC(=O)OOH + O_2 \qquad\qquad (R3b)$$
$$\rightarrow \qquad RC(=O)OH + O_3$$

$$RO_2\cdot + R'O_2\cdot \qquad \rightarrow \qquad RHC=O, \ ROH, \ RC(=O)OH\cdot \qquad (R4a)$$
$$RO_2\cdot + R'O_2\cdot \qquad \rightarrow \qquad RO\cdot \qquad\qquad (R4b)$$

As observed in several studies with highly oxygenated multifunctional organic molecules (HOM, e.g. Berndt et al., 2018a, b; Ehn et al., 2012; Ehn et al., 2014; Mentel et al., 2015) accretion products can be formed in peroxy - peroxy radical reactions:

$$RO_2\cdot + R'O_2 \qquad \rightarrow \quad ROOR' + O_2 \qquad\qquad (R5)$$

In laboratory studies HOM accretion products can contribute significantly to SOA yields (McFiggans et al., 2019).

The presence of NO$_X$ opens new pathways with large reaction rates and production of organic nitrates (R6a), including PAN-like compounds (R7). Furthermore, substantial amounts of alkoxy radicals (R6b) are formed:

$$RO_2\cdot + NO \qquad \rightarrow \qquad RONO_2 \qquad\qquad (R6a)$$
$$RO_2\cdot + NO \qquad \rightarrow \qquad RO\cdot + NO_2 \qquad\qquad (R6b)$$
$$RC(=O)O_2\cdot + NO_2 + M \qquad \leftrightarrow \qquad RC(=O)O_2NO_2 + M \qquad (R7)$$

From reactions R3 to R7 it is obvious that the chemically stable products of peroxy radicals will have different termination groups under low and high $NO_X$ conditions: while hydroperoxides, ketones, carboxylic acids etc. predominate at low $NO_X$ conditions, organic nitrates (ON), including PAN like compounds become more important at high $NO_X$ conditions.

Recent studies demonstrated the dominant role of HOM in SOA mass formation (Ehn et al., 2014; Jokinen et al., 2015; McFiggans et al., 2019; Mutzel et al., 2015; Zhang et al., 2017). HOM are formed by addition of molecular oxygen to alkyl radicals that are formed after H-migration in peroxy- or in alkoxy radicals. Due to a relative long lifetime of peroxy radicals such H-shifts with addition of molecular oxygen can appear several times in sequential steps and is therefore termed autoxidation (Crounse et al., 2011; Mentel et al., 2015; Rissanen et al., 2014). This process leads to highly oxygenated multifunctional peroxy radicals (HOM peroxy radicals). If the respective HOM is formed exclusively via autoxidation of peroxy radicals, the HOM moieties are very likely multiple hydroperoxides (Berndt et al., 2016; Rissanen et al., 2014). If there are intermediate steps via alkoxy radicals there maybe also alcohol groups (Mentel et al., 2015). Bianchi et al. (2019) suggested using the notation HOM when the autoxidation products carry six or more O-atoms.

HOM are low volatile organic compounds (LVOC) or even extremely low volatile organic compounds (ELVOC) and they substantially contribute to mass formation of particles and support NPF (Bianchi et al., 2016; Ehn et al., 2014; Kirkby et al., 2016; Lehtipalo et al., 2018; Tröstl et al., 2016). All experimental evidence show that HOM peroxy radicals terminate with similar rates as less functionalized peroxy radicals (Berndt et al., 2016; Bianchi et al., 2019; Ehn et al., 2017). At low $NO_X$ levels, HOM hydroperoxides, HOM alcohols, HOM carboxylic acids and HOM percarboxylic acids as well as HOM ketones are expected from the termination step and in addition HOM accretion products (HOM-ACC).

At high $NO_X$ levels HOM-ON become important termination products. In addition HOM-ACC products are suppressed (Lehtipalo et al., 2018, Rissanen, 2018) and shifted to smaller C-numbers. The effect of $NO_X$ on HOM-$RO_2$ chemistry is important for understanding of impact of $NO_X$ on SOA formation. In this paper we analyze two aspects of the effect of $NO_X$ which are important for SOA yield: the formation and the volatility of HOM-ON as well as the suppression of HOM accretion products.

## 2 Experimental

### 2.1 Description of the chamber setup and experiments

Experiments were conducted in the Jülich Plant Atmosphere Chamber (JPAC, Mentel et al., 2009; 2013). The actual setup of the chambers was already described in several recent publications (Ehn et al., 2014; Mentel et al., 2015; Sarrafzadeh et al., 2016; Wildt et al., 2014). A 1.45 $m^3$ chamber made of borosilicate glass and set up in a climate-controlled housing was used for these experiments. The chamber was operated as a continuously stirred tank reactor. About thirty-one liters of purified air per minute were pumped through the chamber resulting in a residence time of approximately 46 min. Mixing was ensured by a fan and the mixing time was about 2 minutes. The inlet flow was provided by two about equal purified air streams with one of them containing ozone and water vapor. In the second stream the VOC of interest was introduced from a diffusion

source. Temperature (16 ± 1 °C) and relative humidity (63 ± 2%) inside the chamber were held constant over the course of the experiments.

The chamber was equipped with several lamps. Two discharge lamps (HQI400 W/D, Osram) served to simulate the solar light spectrum. Twelve UVA discharge lamps (Philips, TL 60 W/10-R, 60W, $\lambda_{max}$ =365 nm) provided the photolysis of $NO_2$ with a photolysis frequency $J(NO_2)$ of ~$3.3\times10^{-3}$ $s^{-1}$. An UVC lamp (Philips, TUV 40W, $\lambda_{max}$ =254 nm) was used to produce OH radicals by ozone photolysis and reaction of $O^1D$ atoms with water vapor. This lamp was housed in a quartz tube across the chamber diameter and parts of the lamp were shielded by movable glass tubes. By altering the gap between these glass tubes, the photolysis frequency for ozone, $J(O^1D)$ and therewith the OH production rate could be adjusted. During most of the experiments described here, $J(O^1D)$ was about $2.9\times10^{-3}$ $s^{-1}$. The photolysis frequencies $J(NO_2)$ and $J(O^1D)$ (as function of the TUV gap) were determined experimentally by actinometry with $NO_2$ and $O_3$ in the chamber. Gas-phase compounds were measured such as ozone ($O_3$, UV-absorption, Thermo Environmental 49), nitrogen monoxide (NO, chemiluminescence, Eco Physics, CLD 770 AL ppt), and nitrogen dioxide ($NO_2$, chemiluminescence after photolysis, Eco Physics, PLC 760). Water vapor concentrations were measured by dew point mirror (TP- 2, Walz).

We used the monoterpenes α-pinene and β-pinene (both Aldrich, 95% purity) as SOA precursors. The monoterpenes (MT) were measured either by gas chromatography mass spectrometry (GC-MS; Agilent GC-MSD system with HP6890 GC and 5973 MSD) or by proton transfer reaction mass spectrometry (PTR-MS; Ionicon, Innsbruck, Austria). Both devices were switched between the outlet and the inlet of the chamber in order to quantify concentrations in the chamber and to determine the VOC source strength.

To provide $NO_X$ to the photochemical system we added $NO_2$ (Linde, 100 ppm $NO_2$ in nitrogen) in the *β*-pinene experiments or NO (90 ppm in nitrogen) in the *α*-pinene experiments to the VOC containing air stream. In case of $NO_2$ addition, a fraction of the added $NO_2$ was converted to NO due to the $NO_2$ photolysis. In case of NO addition, the major portion of the added NO was converted to $NO_2$ by reaction with $O_3$. We adjusted the $O_3$ addition such that a steady state $[O_3]$ within a range of 60 - 90 ppb was achieved. We observed memory effects for $NO_X$ probably due to Teflon parts (tubing, fan) in the chamber. In particular after experiments at high $NO_X$ concentrations residual $NO_X$ appeared in the chamber on the next day when switching on the TUV lamp. To minimize the memory effect, the chamber was operated without $NO_X$ for one day between the $NO_X$ experiments. The background $NO_X$ concentration was around 300 ppt. When adding $NO_X$, its initial concentration, $[NO_X]_0$ was between 5 and 150 ppb and thus substantially above the background levels.

The results presented here were obtained at steady state conditions when all physical and chemical parameters were constant or in steady state, respectively. To indicate reference to steady state we mark the concentrations of MT and $NO_X$ with the subscript "SS". To allow better comparison to literature data we refer in some instances to the initial concentrations $[NO_X]_0$, [α-pinene]$_0$, and [β-pinene]$_0$, indicated by subscript "0", which are in our case input concentrations.

The OH concentrations were calculated from the decay of the respective MT in the chamber (Eq.2).

$$\frac{d[VOC]}{dt} = \frac{F}{V}([VOC]_0 - [VOC]) - (k_{OH} \cdot [OH] + k_{O3} \cdot [O_3]) \cdot [VOC] \qquad (1)$$

$$[OH]_{SS} = \frac{\frac{F}{V} \cdot \frac{[VOC]_0 - [VOC]_{SS}}{[VOC]_{SS}} - k_{O3} \cdot [O_3]_{SS}}{k_{OH}} \qquad (2)$$

Equation (1) describes mass balance of the MT in the chamber and Eq. (2) results from Eq. (1) under steady state conditions when d[VOC]/dt = 0. In Eq. (1) and (2), V is the volume of the chamber, F is the total air flow through the chamber, $[VOC]_0$ and [VOC] are the concentrations of the VOC under investigation in the inlet air and in the chamber, respectively. $k_{OH}$ and $k_{O3}$ are the respective rate coefficients for the reactions of the VOC with OH and with $O_3$. In one case, where the concentration of β-pinene was altered during the experiment, m-xylene was added as a tracer for [OH].

Rate constants used for the determinations of [OH] were: $k_{OH} = 5.37 \times 10^{-11}$, $7.89 \times 10^{-11}$, and $2.3 \times 10^{-11}$ cm$^{-3}$ s$^{-1}$ for α-pinene, β-pinene, and m-xylene, respectively (Atkinson, 1994; Atkinson, 1997). At typical conditions with $[O_3]$ ~ 60-100 ppb and [OH] ~ $3 \times 10^7$ the consumption of α-pinene by $O_3$ ($k_{O3} = 8.7 \times 10^{-17}$ cm$^3$ s$^{-1}$) was about 5% compared to that by OH. α-pinene losses due to ozonolysis were therefore neglected for the determination of [OH]. As β-pinene has an even lower ozonolysis rate constant ($k_{O3} = 1.5 \times 10^{-17}$ cm$^3$ s$^{-1}$, Atkinson, 1997), and m-xylene does not react with $O_3$ ($k_{O3} < 1.5 \times 10^{-21}$ cm$^3$ s$^{-1}$, Atkinson

et al., 1994) ozonolysis reactions were also neglected for OH estimations using these VOC. The overall uncertainty in OH concentration was estimated to be about 20% (Wildt et al., 2014).

Our results were obtained in different types of experiments. An overview of the performed experiments with their starting conditions is given in Table 1. Details of the procedures applied during individual experiments are described in the respective sections. Three experiment series (1-3) were conducted to characterize gas phase products and experiment series 4

was conducted to characterize the particle phase. In the first experiment we estimated the molar yield of all organic nitrates (ON) independent of their contribution to particle formation. This was made at the example of β-pinene at one $NO_X$ concentration (see section 3.1). Experiment series 2 were performed to determine the fraction of HOM organic nitrates at the total amount of HOM. Both monoterpenes were used with several $NO_X$ levels as indicated (see section 3.3). In experiment series 3 we determined effective uptake coefficients for HOM at a low and at a high $NO_X$ level for HOM-ON and other

HOM termination products (see section 3.4). These experiments were performed with seed particles (ammonium sulfate), which were dried by a silica gel diffusion dryer to RH < 30% before they entered the chamber. In the last experiment series 4 we characterised the amount of nitrate bound to organics in particles for comparison with the amount of organic bound nitrate in those HOM-ON that efficiently contribute to particle formation.

**2.2 Determination of highly oxygenated molecules (HOM)**

Highly Oxygenated Molecules (HOM) were measured by a Chemical Ionisation Mass Spectrometer operated with nitrate as reagent ion ($NO_3^-$-CIMS, Jokinen et al., 2012). First we determined that the relative transmission curve of our $NO_3^-$-CIMS

was flat (supplement section S1.1). This indicates that detection of HOM within the mass range from ~230 Da to ~600 Da is nearly mass independent. So far, no absolute calibration method exists for HOM. However, the charging efficiency for HOM is near to the kinetic limit like that for sulfuric acid (Ehn et al., 2014; Kirkby et al., 2016). Thus, the sensitivity of the $NO_3^-$-CIMS to HOM is supposedly similar to that of sulfuric acid. We therefore calibrated our $NO_3^-$-CIMS to sulfuric acid and used the calibration coefficient for the HOM, too (see supplement section S1.2). Applying the sulfuric acid calibration coefficient to the HOM we investigated the mass balance between condensable HOM and formed particle mass which was closed within a factor of 2 (see supplement section S1.3). It is also likely that the sensitivity of the $NO_3^-$-CIMS does not depend much on the functional group formed in the final termination step. This assumption is reasonable as HOM formed by autoxidation contain several hydroperoxy groups or a mixture of hydroxy and hydroperoxy groups, if they are formed via the alkoxy peroxy path (Mentel et al., 2015). Furthermore, a good agreement of the fraction of organic nitrates on the total reaction products of β-pinene (see section 3.1) and the fraction of HOM-ON of the total HOM gives us confidence that the uncertainty induced by this assumption does not affect much the main conclusions.

This all together with the quantum mechanical results of same sensitivity for HOM containing 6 or more O-atoms (Hyttinen et al., 2017) gave us confidence that concentrations of HOM with 6 or more O atoms were determined with the same sensitivity to an uncertainty of less than a factor of two.

In summary, we used the calibration coefficient for $H_2SO_4$ to calculate HOM concentrations. We further concluded a same sensitivity for the detection of all HOM including accretion products when they contained 6 and more O-atoms. These conclusions are supported by observations of Breitenlechner et al. (2017) who found that, once the HOM contains more than 5 O-atoms, the sensitivity is to a good approximation independent of the number of O-atoms. The sensitivity of the $NO_3^-$-CIMS is unclear for compounds containing less than 5 O-atoms (Hyttinen et al., 2017; Rissanen et al., 2014; Riva et al., 2019). However, as will be shown in section 3.4, HOM with less than 5 O atoms are of minor importance for particle mass formation, hence we will neglect them and this will not contribute much uncertainty to our results.

Identification of molecular formulas for individual HOM was obtained using high resolution spectra (resolution power ≈4000) as described in the supplement section S2. In case of HOM spectra from β-pinene photo-oxidation we found many not-fully-resolved double peaks from the overlapping of $C_{10}$, $C_9$, and $C_8$ progressions. The mass spectra of α-pinene HOM in general consisted of singular peaks, i.e. they were quite well resolved. Figure S5 in supplement section S2 shows how HOM-$RO_2$ and HOM-ON were separated with increasing [$NO_X$]. In the high resolution analysis of α-pinene we focus on the mass range 230 Da to 550 Da. The lower limit 230 Da was chosen because of the equal sensitivity of the $NO_3^-$-CIMS towards HOM as discussed before and because $C_{10}$ compounds with 6 or more O atoms have molecular weights > 230 Da. As we will show in section 4.2 they are LVOC and ELVOC and will contribute to SOA formation. The upper limit was set to 550 Da to reduce the influence of noise since not much signal is found for molecular masses > 550 Da.

The observed concentration of gas-phase HOM depends on the OH concentration and the condensation sink provided by newly forming particles. Adding to or removing $NO_X$ from a given photochemical system directly impacts [OH] by reactions

R1 and R2. $NO_X$ furthermore suppresses new particle formation (Wildt et al., 2014) which leads to a decreasing condensation sink for HOM with increasing $[NO_X]$. The actual OH concentration affects the actual turnover of the precursor, thus the actual production of $RO_2$· and HOM, while the actual condensational sink leads to condensational loss of HOM. Both factors change the observed HOM gas-phase mixing ratio and can superimpose the impacts of $NO_X$ on peroxy radical chemistry itself. In order to separate the chemical impacts of $NO_X$ on HOM peroxy radical chemistry, we needed to take out the effects of [OH] and condensational sink as much as possible. This was achieved by normalizing the HOM mixing ratio to particle free conditions and to a certain reference oxidation rate. The procedure is described in detail in supplement section S3.

## 2.3 Particle-phase measurements

To characterize the particle phase we used a condensation particle counter (CPC, TSI 3783), a scanning mobility particle sizer (Electrostatic classifier TSI 3080, including a differential mobility analyzer TSI 3081 and a CPC TSI 3025A) and an aerosol mass spectrometer (AMS, Aerodyne HR-ToF-MS, modified for application on a Zeppelin airship, (Rubach, 2013)). In the AMS the aerosol particles were vaporized at 600 ℃ and ionized by electron impact ionization at 70 eV. The AMS was routinely operated in V-mode in two alternating modes: 1 min. MS mode to measure the chemical composition and 2 min. PToF mode. Only MS mode data were analyzed here. In the following we will use the amount of nitrate bound to organics ($OrgNO_3$) as a diagnostic to link observation of HOM-ON in the gas-phase to observations in the particulate phase. We separated organic and inorganic particulate nitrate and determined the amount of $OrgNO_3$ by the $NO_2^+/NO^+$ method for AMS (Farmer et al., 2010; Kiendler-Scharr et al., 2016).

SOA yields were determined as described in Sarrafzadeh et al. (2016). For determining mass yields, the particle mass formed during steady state conditions was divided by the mass of the consumed MT, which is the difference between inlet and outlet concentration:

$$Y = \frac{produced\ particle\ mass}{BVOC\ consumption} \tag{3}$$

During measurements of particle mass, the mean diameter of particles was above 100 nm. As the loss rates of such particles on the chamber walls were low (Mentel et al., 2009), they were neglected. Losses of oxidized SOA precursors to the chamber walls were considered by applying the correction function given by Sarrafzadeh et al. (2016). This function describes the ratio of wall losses over the sum of wall losses and losses on particles. For the data given here, the correction factors were between 1.5 and 2.1.

## 2.4 Determination of effective uptake coefficients

The experiments to determine effective uptake coefficients for HOM, $\gamma_{eff}$, were performed as follows: Signal intensities of the respective HOM were measured at zero (α-pinene) or low particle load (β-pinene). Then we introduced dried seed particles into the reaction chamber by spraying ammonium sulfate solutions in two steps with concentrations of 4g/L and 40 g/L. The particles were dried by passage through a silica gel diffusion tube and size selected at an electromobility diameter d = 100 nm. Increasing amounts of ammonium sulfate seed particles instantaneously led to lowered HOM concentrations in the gas phase due to the additional loss by condensation on the seed particles. The decrease in signal intensity with increasing particle surface was used to evaluate $\gamma_{eff}$ as described below.

We operate our chamber as a continuously stirred tank reactor (CSTR) in a flow through mode with a well-mixed core of the chamber. HOM lost at the chamber walls must diffuse through the laminar boundary layer at the chamber walls. In the chamber with no aerosols present, the walls constitute the major sink of HOM. In this case the observed steady state concentration is determined to a very good approximation only by the production rate and the wall loss rate. When seed aerosol is added or new particles are formed, the additional condensational sink provided by the particle surface lowers the steady state concentration. Under conditions of unperturbed gas-phase production and typical times for phase transfer smaller than the residence time of the air in the chamber, the lowered gas-phase steady state concentrations reflect the partitioning of HOM, which is determined by the balance of condensation and evaporation. (Tröstl et al. (2016) noted that HOM can be LVOC i.e. they have a very small but noticeable vapor pressure.) Since we are working in a steady state system, we cannot easily separate between a kinetically slow uptake and a balance between (fast) uptake in steady state with a (fast) evaporation.

For molecules with noticeable volatility steady state between condensation and evaporation is established on the time scales of less than 10 minutes in our CSTR, e.g. for molecules with molecular masses of 300 Da and at a particle surface of $5.0 \times 10^{-4}$ $m^2$ $m^{-3}$ the typical uptake time is about the same as the mixing time of 120 s. We express the net effect of condensation and evaporation by an effective uptake coefficient $\gamma_{eff}$ and the gas kinetic collision rate of HOM with the particle surface. The $\gamma_{eff}$ can be determined by measurement of the ratio of steady state HOM concentrations for the unseeded case and for seeded cases with the advantage that only signal intensities are required and hence no calibration is needed (Sarrafzadeh et al., 2016).

Size selection by electromobility produced bimodal size distributions in the chamber over the times of observation of 2 h and 3.5h with one mode around 100 nm and a second mode around 200 nm (≈25% by number). The diameter of the median of the surface distributions was located in a range of 150 - 200 nm. Since it is likely that nearly every collision with the surface of particles will lead to phase transfer of HOM, we considered the Fuchs-Sutugin correction factor ($f_{FS}$) to calculate the collision rate (Fuchs and Sutugin, 1971) in order to correct for diffusion limitations. Taking into account the mean free path for a range of molecular compositions of $C_{10}H_{14-16}O_{4-12}$ and a median of the particle surface distribution in a range of 150-200 nm, we estimate $f_{FS}$ in a range of 0.65-0.75; diffusivity was calculated after Fuller et al. (1969).

In a first step wall loss rates of HOM were measured. After stopping the OH production and thus photochemical HOM formation we observe an exponential decay of HOM signals. The exponential decay of the signal intensity gives the lifetimes of those HOM. In absence of particles the lifetimes reflect the wall loss rates. As shown in Sarrafzadeh et al. (2016) and Ehn et al. (2014) the lifetimes were in the range of 70 to 150 s, i.e. the loss rates on the walls of the chamber, $L_W(HOM)$, were in the range of $1.4 \times 10^{-3}$ to $7 \times 10^{-3}$ s$^{-1}$. $L_W(HOM)$ are more than order of magnitude higher than those caused by the flush out of the air in the chamber ($3.6 \times 10^{-4}$ s$^{-1}$ for the residence time of 46 minutes). Therefore we neglected flush out as sink for HOM. In a second step, the chemical system was kept at the same steady state conditions for $[OH]_{SS}$, $[O_3]_{SS}$, and MT concentration. Data evaluation was based on the following considerations: The concentration of any HOM, c(HOM) is determined by its production rate P(HOM) and the first order loss rate, L(HOM) as given in Eq. (4):

$$c(HOM) = \frac{P(HOM)}{L(HOM)} \tag{4}$$

In absence of particles the total loss rate L(HOM) is given alone by the loss rate at the chamber walls, $L_W(HOM)$. In presence of particles L(HOM) is the sum of loss rates at the walls and at the particle surface, $L_W(HOM) + L_p(HOM)$. At constant production rate P(HOM) the ratio of concentrations is inverse proportional to their ratio of loss rates:

$$\frac{c(HOM)^0}{c(HOM)} = \frac{L_W(HOM) + L_P(HOM)}{L_W(HOM)} \tag{5}$$

In Eq. (5), $c(HOM)^0$ is the concentration of HOM in the particle-free chamber and c(HOM) is the concentration in the particle-containing chamber. Solving Eq. (5) for $L_p(HOM)$ we achieve Eq. (6):

$$L_P(HOM) = \frac{c(HOM)^0}{c(HOM)} \cdot L_W(HOM) - L_W(HOM) \tag{6}$$

We varied the surface area of seed particles ($S_P$) and determined $L_P(HOM)$ by Eq.(6). We found a linear relationship between $L_P(HOM)$ and $S_P$ as expected from kinetic gas theory, Eq. (7):

$$L_P(HOM) = \gamma_{eff} \cdot f_{FS} \cdot \frac{\bar{v}}{4} \cdot S_P \tag{7}$$

In Eq. (7), $f_{FS}$ is the Fuchs-Sutugin correction, $\bar{v}$ is the mean molecular velocity of the HOM, and $\gamma_{eff}$ is an effective uptake coefficient. The coefficient $\gamma_{eff}$ was obtained from the slope of such plots by dividing the values for slopes by $f_{FS} \times \bar{v}/4$ with $f_{FS} = 0.7$ assuming a mean median of the surface size distribution of 175 nm. In case of β-pinene some new particle

formation was observed which hindered measuring $c(HOM)^0$ directly. Here, $c(HOM)^0$ was calculated by linear extrapolation of $1/c(HOM)$ to zero particle surface (compare to supplement, Section S3).

It has to be noted that $\gamma_{eff}$ is only valid if $S_P$ is not too large for two reasons. First, in presence of a strong condensational sink, many HOM signals become close to the detection limit (here for $S_p > 1.2 \times 10^{-3}$ $m^2$ $m^{-3}$). Secondly, for large $S_p$ ($>2 \times 10^{-3}$ $m^2$ $m^{-3}$) distinct deviations from linearity were observed, likely due to the fact that the time scales of losses of peroxy radicals on particles become similar to the time scales of peroxy radical reactions (Pullinen, 2017). If so, the production rates P(HOM) of HOM termination products are not constant but decrease significantly with increasing particle load.

# 3 Results

## 3.1 Yields of organic nitrates from β-pinene photo-oxidation

In the first step we determined the potential of organic nitrate (ON) formation in a β-pinene / $NO_X$ mixture ([β-pinene]$_0$ ~39 ppb / [$NO_X$]$_0$ ~50 ppb). The β-pinene and $NO_X$ were added to the chamber that contained about 60 ppb $O_3$. The OH production was started by switching on the UV lamp (time = -2.4h in Figure 1), inducing the photochemical oxidation of β-pinene and thereby the ON production. As shown in Figure 1, concentrations of β-pinene and $NO_X$ decreased in the presence of OH. When the photochemical system was in a steady state after about two hours (time t = 0 h in Figure 1) the β-pinene addition was stopped and β-pinene concentration decreased to zero. In parallel, [OH] increased leading to a lower $NO_X$ concentration. At time t = 1.7 h, the OH concentration was re-adjusted to the same OH level as before the removal of β-pinene by lowering $J(O^1D)$. The decrease in [OH] caused an increase of [$NO_X$] by 15 ppb to 32 ppb. Considering the $NO_X$ level of 20 ppb before the β-pinene had been removed, the net [$NO_X$] increase amounts to 12 ppb. The inflow of $NO_X$ as well as the OH concentration were the same before and after removal of β-pinene, but now [$NO_X$]$_{SS}$ was higher. Hence, with β-pinene we removed a strong $NO_X$ sink in the chamber. Most of this $NO_X$ sink is made up by reactions of NO and $NO_2$ with peroxy radicals and peroxy acyl radicals that lead to ON formation (reactions R6a and R7). Thus, the difference in [$NO_X$]$_{SS}$ in presence and in absence of β-pinene allowed to calculate the fraction of ON formed from β-pinene. Defining the yield of ON formation as the molar amount of $NO_X$ "released" by not forming β-pinene ON over the molar amount of consumed β-pinene and with the assumption that one lost $NO_X$ molecule had produced one ON molecule we derived a molar yield of ~36% for the ON formed from β-pinene. For later comparison with AMS results we calculated the mass concentration of nitrate bound to the organic moieties (OrgNO$_3$), again with the assumption that one lost $NO_X$ molecule produces one OrgNO$_3$. A total mass concentration of 33 μg m$^{-3}$ OrgNO$_3$ in the gas phase was obtained at the given condition in the chamber. The mass concentration of HNO$_3$ formed during this time was about 24 μg m$^{-3}$ (for details of these calculations see supplement section S3).

## 3.2 HOM formation from α-pinene- and β-pinene photo-oxidation

We observed multifunctional peroxy radicals (HOM-RO$_2$) as well as their termination products in the high resolution mass spectra. The latter are formed in accordance with established pathways of peroxy radical chemistry (Bianchi et al., 2019). We will distinguish HOM-PP, which arise from permutation reactions of HOM-RO$_2$ with peroxy radicals including HO$_2$, and HOM-ON, which are formed in reaction of HOM peroxy and HOM acyl peroxy radicals with NO or NO$_2$. In addition we found HOM accretion product with C$_{>10}$ and C$_{<20}$ (HOM-ACC).

We observed two major differences in HOM formation and product patterns for α-pinene and β-pinene, which are both related to the position of their double bond:

1. For $α$-pinene, with an endocyclic double bond, addition of ozone is relatively fast and at OH concentrations up to $1\times10^7$ cm$^{-3}$ certain fractions of HOM were produced by ozonolysis (compare supplement Figure S6). In contrast, ozonolysis of β-pinene, with an exocyclic bond, is slow and does not produce significant amounts of HOM (Pullinen, 2017).

2. The HOM products of α-pinene in the monomer region mainly consisted of C$_{10}$-molecules, as breaking of an endocyclic double bound will merely lead to ring opening. The breakage of the exocyclic double bond of β-pinene during the oxidation process will cause some fragmentation. We observed progressions of C$_{10}$, C$_9$, C$_8$, and C$_7$ HOM leading to overlapping peaks in the mass spectra. E.g. molecular masses where 1 C-atom and 4 H-atoms are replaced by 1 O-atom were not fully resolved. However, by peak fitting we were able to attribute the contributing formula components in most cases (see supplement section S2 and peak list in supplement section S6). In addition, fragmented peroxy radicals in β-pinene form a larger variety of accretion products with less than 20 C-atoms.

Besides these differences, the behavior with respect to NO$_X$ addition was very similar for α-pinene and β-pinene photo-oxidation: increase of HOM-ON (increase of peaks with odd molecular masses), decrease of accretion products and a shift of HOM monomers to the higher $m/z$.

## 3.3 Accretion products and products from fragmentation

With increasing [NO$_X$] we observed a strong decrease of HOM-ACC relative to HOM monomers. These can be clearly seen in the HOM mass spectra obtained for α-pinene and β-pinene in Figure 2 by comparing the ranges of $m/z < 340$ Da and $m/z > 420$ Da for low and high NO$_X$ conditions. To quantify the effect, the molecular mass weighted signals of HOM-monomers (C$_5$-C$_{10}$, 230 Da $< m/z < 550$ Da) and HOM-ACC products (C$_{11}$-C$_{20}$, 230 Da $< m/z < 550$ Da were converted to mass concentrations, summed up and normalized as described in supplement section S3. Figure 3 shows the mass concentration of total HOM and the fractions of HOM monomers and HOM-ACC as a function of [NO$_X$]. HOM-ACC decrease with [NO$_X$]: at the lowest and highest NO$_X$ levels of 0.3 ppb and 72 ppb HOM-ACC contribute 0.3 μg m$^{-3}$ and 0.09 μg m$^{-3}$, respectively,

to total HOM, whereas HOM monomers contribute about 0.4 µg m$^{-3}$ over the whole range. At low $NO_X$ conditions, accretion products contributed ≈40 % and monomers contributed ≈60 % to the total mass concentration of HOM. At the highest $NO_X$ level the mass concentration of HOM-ACC was suppressed by about 70%. In comparison, the sum of HOM monomers was diminished only by less than 10 %. The decrease of HOM-ACC mixing ratio is attributed to the competition between HOM-

5 ON formation channels (R6a and R7) and HOM-ACC formation channel (R5). As we will show in the next section, in presence of $NO_X$ more HOM-ON monomers were formed and thus less HOM accretion products.

We separated HOM monomers in compounds with $C_{10}$ and compounds with $C_{<10}$ ($C_{5-9}$) shown by the small circles in Figure 3. The mass concentration of $C_{<10}$ HOM increased with $[NO_X]_{SS}$ (small open circles) whereas $C_{10}$ HOM decreased (small grey circles). Compounds with $C_{5-9}$ double from ≈0.9 to ≈1.8 µg m$^{-3}$ at the highest $[NO_X]_{SS}$, whereas the $C_{10}$ compounds drop

by only about 30%. Since $C_{5-9}$ compounds arise in large parts from fragmentation of alkoxy radicals, this indicated that a portion of $C_{<10}$ arose from fragmentation of alkoxy radicals formed in R6b. Since we consider only molecular masses ≥ 230 Da, which is the molecular mass for formula $C_{10}H_{14}O_6$, the $C_{<10}$ compounds must still be highly functionalized, i.e. they must carry more oxygen than the respective $C_{10}$ compounds to reach similar molecular masses, actually one O atom more per C lost.

We assume that compounds in the selected mass range will contribute to SOA formation. Since total HOM decrease at higher $[NO_X]_{SS}$, but HOM monomers remain about stable over the whole $NO_X$ range, the suppression of the HOM-ACC were the cause for the reduction of total HOM and therewith of condensable mass. The fragmentation via alkoxy radicals played only a minor role.

### 3.4 Detection of termination permutation products of peroxy radical - peroxy radical reactions and HOM-organic
nitrates

At low $NO_X$ conditions, HOM-PP with even molecular masses showed the highest concentrations. HOM-ACC are also produced from peroxy-peroxy permutation reactions, but their intensity decreased strongly and HOM-ACC are barely observed at high $NO_X$ condition, as described in previous section 3.3. For that reason, we focus on monomer products in the following considerations in section 3.5-3.7.

Because of the complexity of the product spectrum the character of the functional group formed in the termination step of HOM-PP cannot be derived unambiguously from the available elementary molecular formulas alone. As an example, HOM-PP with the molecular formula $C_{10}H_{16}O_X$ can be hydroperoxides formed from peroxy radicals with molecular formula $C_{10}H_{15}O_X$ in reaction R3a, they can be alcohols formed from peroxy radicals $C_{10}H_{15}O_{X+1}$ in reaction R4a, or they can be ketones formed from peroxy radicals with the molecular formula $C_{10}H_{17}O_{X+1}$ in reaction R4a. Dependent on the specific

precursor peroxy radical, they can also be carboxylic acids or percarboxylic acids. We therefore lump the monomer HOM with even masses (HOM-PP) together independent of the chemical character of the termination group. In case of β-pinene, we did not separate the contributions from different progressions (e.g. $C_{10}H_YO_X$ and $C_9H_{Y-4}O_{X+1}$) but used the overall signal under the peak for further analysis and indicate the main components under the peak (e.g. Figure. 5). Separation between

HOM-PP and HOM-ON was easier, because HOM-ON have odd molecular masses. Peroxy radicals (without N) also contribute to the odd mass peaks, but in most cases they could be separated by the mass defect. Other contributors to odd mass peaks are HOM containing $^{13}C$ and clusters with the nitrate dimer ($HNO_3NO_3^-$) of HOM-PP, but the latter contribution was small (supplement section S2). Analysis of the high resolution mass spectra for $\alpha$-pinene revealed that HOM-ON-peroxy

radicals are rare. Peak lists for the HOM measured in the absence and presence of $NO_X$ are given in the supplement section S6. (Note that we cannot *a priori* distinguish between HOM-ON formed in reactions R6a and R7, respectively.)

Figure 4 demonstrates the change in the HOM pattern when $NO_X$ is added to the reaction system. At low $NO_X$ levels, HOM-PP (black) were predominant, but already at the medium $NO_X$ levels the concentrations of HOM-ON (blue) were similar to HOM-PP (black). HOM-ON concentrations increased on the cost of HOM-PP. The product spectra of $\beta$-pinene showed a

similar shift from HOM-PP to HOM-ON.

With increasing $NO_X$ we observed a small but increasing fraction of highly oxidized nitrates with C<10. Their chemical formulas have low H:C ratio compared to gas-phase $C_{10}$ HOM-ON. Supposedly, they did not arise from gas-phase chemistry but were likely formed at the walls. Their time series was not responding to the changes of experimental conditions like start of photochemistry or cease of $NO_X$ addition, instead these compounds increased steadily after $NO_X$ addition. For $\alpha$-pinene,

their maximum contribution appeared at 74 ppb $[NO_X]_{SS}$ where they mounted up to 8% of the total HOM concentration and 17% of the HOM-ON concentration. Below 35 ppb $NO_X$ their contribution was less than 7% and 12%, respectively. For $\beta$-pinene, this was less distinct and the contribution was only 2 to 3 %. Because these HOM-ON had C<10 and appear at the lower end of the mass spectrum, their contribution to HOM mass (and therewith SOA mass) were small, and we did not correct for these products.

**3.5 Effective uptake coefficients for HOM-PP and HOM-ON.**

Based on identified HOM-ON and HOM-PP, we characterized their potential contribution to SOA formation by determining their loss rates on seed particles $L_P$(HOM), Eq. (6). According to Eq. (7), plots of $L_P$(HOM) versus particle surface, $S_P$, should exhibit a linear dependence between $L_P$(HOM) and $S_P$ allowing for determining effective uptake coefficients, $\gamma_{eff}$. However, for conditions with sufficient HOM production, we could not fully suppress new particle formation in absence of

seed particles. As a result $c(HOM)^0$ in Eq. (5) and Eq. (6) had to be determined by extrapolating $1/c$(HOM) as $f(S_P)$ to $S_P$=0. The wall loss coefficient $L_W$(HOM) was determined to 150s in independent experiments at lower concentrations and in absence of new particle formation. With $c(HOM)^0$ and $L_W$(HOM) $L_P$(HOM) was calculated allowing to derive $\gamma_{eff}$ from Eq. (7) by applying $f_{FS}$ = 0.7 to the slope of $L_P$(HOM) as a function particle surface $S_P$ (Figure 5). Figure 5 is based on HR data, which show substantial scatter. The UMR data showed a better signal to noise ratio than the individual HR peaks under the

same UMR signal. In order to reduce the scatter we thus used the respective unit mass resolution (UMR) data for the evaluation of $\gamma_{eff}$. In Figure 6 we compare $\gamma_{eff}$ for HOM with the same number of O-atoms. Note that compared to their chemical sum formula, ON were shifted by one O to lower O in order to account for the addition of NO. When comparing data for HOM with the same numbers of O-atoms (in the precursor peroxy moiety), no significant and systematic differences

were found for $\gamma_{eff}$ within the uncertainty limits, i.e. the potential to condense on particles was about the same for HOM-PP and HOM-ON. For HOM moieties with 8 and more O-atoms, $\gamma_{eff}$ approaches 1 independent if they were HOM-PP or HOM-ON. HOM-PP and HOM-ON with 6 and more O-atoms with upper limits of $\gamma_{eff}$ near 0.5 will still reside to a large portion in the particle phase therefore they should also contribute significantly to SOA mass.

## 3.6 Organic bound nitrate in SOA and in gas-phase HOM-ON

SOA yields for α-pinene and β-pinene (wall loss corrected) were between 0.08 and 0.18, thus in the same range as those reported by Sarrafzadeh et al. (2016). The amount of formed SOA mass and the SOA yields changed with addition of $NO_X$ to the photochemical system because $NO_X$ affects $[OH]_{SS}$ and thereby also the formation of higher generation products. The mass fractions of $OrgNO_3$ in the SOA particles are shown in Figure 7. They were calculated for the different $NO_X$ concentrations by dividing the mass concentration of $OrgNO_3$ by the mass concentration of the respective organic as measured with the AMS. The fraction of $OrgNO_3$ in particles was dependent on the $NO_X$ concentrations. It was negligible when no $NO_X$ was added and increased steadily with increasing $[NO_X]$. At the same time, also the fraction of inorganic nitrate increased with increasing $[NO_X]$, but was a factor of about 3 lower than that of $OrgNO_3$ (Figure 7). Calculating $[HNO_3]_{SS}$ in the gas phase from $[NO_2]$ and $[OH]$ shows that, at highest NOx concentrations up to 24 µg m$^{-3}$ $HNO_3$ were formed in the gas phase, but less than 0.1 µg m$^{-3}$ of inorganic nitrate was found in the particle phase. For the same $[NO_X]_{SS}$, the mass fractions of organic or inorganic nitrate in SOA were about the same for α-pinene and β-pinene, indicating that the formation of condensable $OrgNO_3$ was similar. The determination of $OrgNO_3$ comprises some uncertainty, but even if we count all inorganic nitrate as $OrgNO_3$ we get an upper limit of less than 4%.

We compared the amount of $OrgNO_3$ in particles to that in the gas phase. As shown in section 3.1, at $[NO_X]_{SS}$ ~20 ppb about 33 µg m$^{-3}$ $OrgNO_3$ was formed in the gas phase. At similar $[NO_X]_{SS}$ and similar β-pinene concentrations ($[NO_X]_{SS}$ ~22 ppb, $[β-pinene]_{SS}$ ~6 ppb) the fraction of $OrgNO_3$ in the particle phase was only 1.6 ± 0.64 % (Fig. 7), i.e. less than 0.4 µg m$^{-3}$ $OrgNO_3$ was bound in particles. We conclude that many ON are too volatile to significantly contribute to the particulate phase. However, considering the low volatility HOM-ON (section 3.5), HOM-ON should contribute to particle mass eventually providing the particulate $OrgNO_3$. We estimated the mass fraction of $OrgNO_3$ bound in HOM-ON. For this we considered all HOM with 6 and more O-atoms in the HOM-moiety (molecular mass >230 Da) because their $\gamma_{eff}$ is large enough to partition significantly into the particle phase and to contribute efficiently to SOA mass. As shown in Figure 6, the HOM partitioned without preference, independent of being HOM-PP or HOM-ON. We first determined the fraction on a molecular base by using the ratios of signal intensities, which is identical to using concentrations c:

$$\frac{c(HOM-ON)}{c(all\ HOM)} \approx \frac{\sum_{241}^{405} c(HOM-ON)}{\sum_{230}^{550} c(allHOM)} \tag{8}$$

In Eq. (8), the left hand term represents the molar fraction of HOM-ON. We summed all HOM with O ≥ 6, which included all HOM with $\gamma_{eff}$ >0.5 and provides a lower limit. Signals at m/z>550 Da were not taken into account, since they were very low at high NO$_X$ levels and thus uncertain. In a second step we calculated from Eq. (8) the mass ratio of OrgNO$_3$. We split all HOM in the denominator of Eq. (8) in HOM-ON and other termination products and multiplied the concentrations with the respective molar weight (Eq. (9)). The numerator was multiplied with the molecular weight of the nitrate termination group:

$$\frac{M(OrgNO_3)}{M(organic)} \approx \frac{\sum_{241}^{405} c(HOM-ON)\cdot 62}{\sum_{230}^{550} c(other\ term.prod.)\cdot m + \sum_{241}^{405} c((HOM-ON)\cdot(m-62))} \tag{9}$$

In words, the left hand term in in Eq. (9) gives the ratio of the total mass of OrgNO$_3$ over the total organic mass of HOM with$\gamma_{eff}$ > 0.5 (O ≥ 6). This value can be compared with the direct AMS observation of OrgNO$_3$. Figure 8 shows the molecular ratios (calculated by Eq. (8)) and the mass fractions (Eq. (9)) in dependence on NO$_X$. For α-pinene we were able to separate HOM-ON and HOM-RO$_2$ unambiguously (see supplement). For β-pinene we give lower and upper limits of the molecular and mass fractions, because of uncertainties in the HR analysis that were induced by the stronger fragmentation and overlapping progressions of compounds with different number of C but same unit molecular mass. For the lower limit shown in Figure 8, we applied a peak list with all identified signals at low and high NO$_X$ concentrations for fitting, whereas the upper limit was achieved by using the peak list optimized for high NO$_X$ cases. (The reason for the spread can be explained as follows: the approach with the peak list with all identified peaks attributes some HOM-RO$_2$ to the HON-ON signal, independent if the specific HOM-RO$_2$ exist in the chemical system or not, while the approach with the high NO$_X$ peak list has the tendency to falsely attribute HOM-ON to existing HOM-RO$_2$ missing in the high NO$_X$ peak list.)

Both, HOM-ON and OrgNO$_3$ mass fraction increased with [NO$_X$], similarly for both MT. For α-pinene about 40 % of the detected HOM were HOM-ON and more than 10 % of the HOM mass was OrgNO$_3$ once [NO$_X$] was larger than 30 ppb ([BVOC]$_{SS}$/[NO$_X$]$_{SS}$ < 2 ppbC/ppb). For upper limit case of β-pinene we achieved about the same fractions as observed for α-pinene (50 ppb NO$_X$ : [BVOC]$_{SS}$/[NO$_X$]$_{SS}$ < 1.1 ppbC/ppb). Since we considered only HOM that efficiently condense on particles, one would expect that OrgNO$_3$ brought by HOM-ON alone should contribute about 10 % of the SOA mass. This was not the case, as the direct comparison in Figure 8 shows. The maximum contribution of particulate OrgNO$_3$ was about 3 %, i.e. the measured OrgNO$_3$ was a factor of 3 to 4 lower than expected: the OrgNO$_3$ bound in low volatility HOM-ON which could potentially contribute to SOA mass was significantly higher than OrgNO$_3$ directly observed in the particle phase.

## 3.7 Mass concentration of HOM

To estimate the possible effect of hydrolysis of $OrgNO_3$ and re-evaporation of $HNO_3$, potential condensable mass concentration ($c^{mass}$) was derived by weighing the concentration ($c_i^N$) of each $HOM_i$ by its molecular mass ($M_i$) in the range of 230 Da to 550 Da:

$$c^{mass} = \sum_{230}^{550} c_i^N \cdot M_i$$

Herein $c^N$ is the concentration that was corrected according to the method described in supplement section S2. Figure 9 shows the calculated mass concentrations with and without hydrolysis in dependence on $NO_X$. The relative uncertainty limits were estimated to 19%. The uncertainty was estimated from the standard deviation of the data obtained at low $[NO_X]$
conditions (9 measurements). Uncertainties of $[NO_X]_{SS}$ were estimated to be $\pm$ 10%. The uncertainty of absolute concentrations caused by the uncertainty of the calibration factor (see supplement S1) is much higher than the uncertainty limits shown in the Figure 9. However, as the systematic error of the calibration factor is the same for each data point it does not affect the observed trend of only somewhat decreasing mass concentrations of HOM with increasing $NO_X$.

In case of HOM-ON, $c^{mass}$ includes the mass of $OrgNO_3$. According to many studies particulate ON is undergoing
hydrolysis, leading to loss of $HNO_3$ (Bean and Hildebrandt Ruiz, 2016; Boyd et al., 2015; Rindelaub et al. 2015; Takeuchi and Ng, 2019). It is not clear if organic material lost from the particulate phase besides $OrgNO_3$ (Fisher et al., 2016, Takeuchi and Ng, 2019, Zare et al. 2019). The efficiency of the hydrolysis of $OrgNO_3$ depends on RH and particle acidity and several studies report fractions of hydrolyzed particulate ON in a range of 10 - 60% (Bean and Hildebrandt Ruiz, 2016; Boyd et al., 2015; Browne et al., 2013; Rindelaub et al. 2015; Takeuchi and Ng, 2019).
We indicate the resulting SOA mass after considering $OrgNO_3$ loss by hydrolysis and evaporation of $HNO_3$ by a prime as $c^{mass}$' in Figure 9 (details for the calculations of $c^{mass}$' see section 4.3). It is obvious from Figure 9 that the mass concentration of condensable HOM is about 30% lower at the highest $NO_X$ conditions compared to those at low $NO_X$ conditions. It is furthermore evident that the differences between $c^{mass}$ and $c^{mass'}$ are quite low which is showing re-evaporation of $HNO_3$ is of minor importance for explaining the SOA mass suppression with increasing $[NO_X]$ in the system. An explanation must then
be the observed strong decrease of the accretion products with increasing $[NO_X]$ as shown in Figure 2.

## 4 Discussion

### 4.1 Organic nitrates and SOA formation.

Several studies on organic nitrates (ON) or organic bound nitrate ($OrgNO_3$) in SOA refer to reactions of unsaturated volatile organic compounds with $NO_3$ ( Claflin and Ziemann, 2018; Faxon et al., 2018; Fry et al., 2013, 2014; Kiendler-Scharr et al.,

2016; Lee et al., 2016a, Ng et al., 2017 and references therein). The pathway of forming ON by $NO_3$ was negligible in our experiments as we applied quite high light intensity, humidity and also high NO concentrations during our experiments. These experimental conditions inhibited formation of $NO_3$ at relevant concentrations because $NO_3$ was efficiently destroyed by photolysis, by reactions with NO and by scavenging of $N_2O_5$ at the humid surfaces of the chamber walls. The HOM-ON

measured during our experiments were formed in reactions R6a and R7.

There are some studies with respect to the SOA content of ON formed by photo-oxidation (Berkemeier et al., 2016; Lee et al., 2016b; Nozière et al., 1999; Rollins et al., 2010; Takeuchi et al., 2019; Xu et al., 2015b; Zhao et al., 2018) wherein in most cases mass fraction of ON of the total SOA mass is reported. According to literature data, ON produced during photo-oxidation or ozonolysis contribute between 3 % (Lee et al., 2016b) and 40% (Berkemeier et al., 2016) to the total SOA mass.

This compares well with the mass fraction of HOM-ON with molecular masses >230 Da, which varied from 0-50% with increasing $NO_X$ (Figure 4). HOM-ON (>230 Da) provide a measure of the expected contribution of ON to SOA as HOM-ON and all other HOM should condense with the same efficiency as shown by their $\gamma_{eff}$ in Figure 6.

We determined $OrgNO_3$ by the AMS as a diagnostics for ON in the particulate phase. The $OrgNO_3$ mass fractions ranged from 0 % (no $NO_X$ addition) to 2.7 %, which is within the range 0.6 - 8% of most literature data but at the lower end

(Nozière et al., 1999; Rollins et al., 2010; Xu et al., 2015b; Berkemeier et al., 2016; Lee et al., 2016b; Zhao et al., 2018). (When mass fractions of particulate ON were given, we estimated $OrgNO_3$ assuming an average molecular mass of 300 Da and one nitrate group per ON; see supplement section S5). In our study we showed that $OrgNO_3$ depends on the $NO_X$ level ([VOC]/[$NO_X$] level) as expected from established peroxy radical chemistry in presence of $NO_X$. This finding can probably explain the wide range of ON and $OrgNO_3$ fractions reported for SOA formation in presence of $NO_X$. Detailed and

meaningful comparison of our data to those reported in literature requires knowing the [VOC]/[$NO_X$] ratios during SOA formation in the respective experiments. The [VOC]/[$NO_X$] ratio is known for experiments made by us in the SAPHIR chamber in Jülich (Zhao et al., 2018). Zhao et al. (2018) achieved a mass fraction of 11% for $OrgNO_3$ which is significantly higher than the $OrgNO_3$ mass fractions in this study. Interestingly, the 11% found by Zhao et al. (2018) are close to the 10% mass fraction of $OrgNO_3$ determined for the total HOM in this study. These findings will be further discussed in section 4.3.

**4.2 Effective uptake coefficients**

We provided data on effective uptake coefficients, $\gamma_{eff}$ , which allowed differentiation between SVOC, LVOC and ELVOC. The dependence of $\gamma_{eff}$ on the number of O-atoms in the HOM (Figure 6) suggests that the $OrgNO_3$ found in SOA predominantly originates from HOM-ON with 6 and more O-atoms (without -NO). We conclude that ON with less than 5 O-atoms are not so important for the formation of SOA at least at atmospheric loads of SOA < 10 µg m$^{-3}$. This conclusion is

confirmed by our observation of small mass fractions of $OrgNO_3$ in HOM monomers (0-10%) and in the particle phase (0-3%) despite of the large ON fractions produced overall in the gas phase (molecular yield > 30%, Figure 1 in section 3.1) . From all ON only a few percent - the HOM-ON - made it into the particulate phase (sections 3.1 and 3.6).

Our findings are in agreement with observations by Lee et al. (2016b) in a field study. They also show that the distribution of signal intensities for HOM-ON in the gas phase is different from that in the particle phase. Comparing the signal intensities for HOM-ON with the same number of O-atoms between gas phase and particle phase, respectively (Figure 2 in Lee et al., 2016b), it seems that the higher the number of O-atoms, the more the ON partition in the particle phase. As there was no calibration for gas phase HOM-ON, absolute numbers for partitioning coefficients or effective uptake coefficients were not obtained. Our data are qualitatively consistent with those of Lee et al. (2016b), suggesting that the basics of HOM-ON condensation in our laboratory studies are similar to those in the environment.

Comparison between $\gamma_{eff}$ determined for HOM-PP and HOM-ON indicated that there were no significant and systematic differences at least for the HOM moieties with more than 6 O-atoms. The variability in $\gamma_{eff}$ in Figure 6 is probably caused by the fact that more than one compound with different structure and functionalization is contributing to each data point. E.g. $C_{10}H_{16}O_x$ showed a large $\gamma_{eff}$ at O number of 7 (Figure 6). The signal assigned to $C_{10}H_{16}O_7$ could be dominated by a compound with extreme low vapor pressure.

The independence of the overall uptake behavior from the termination groups can be understood from the basics of group contribution models (Capouet and Müller, 2006; Pankow and Asher, 2008; Compernolle et al., 2011): HOM peroxy radicals with more than 6 O-atoms already carry protic functional groups - OH, -OOH, C(=O)OH, or C(=O)OOH - from the several autoxidation steps. Thus their vapor pressure is low because of the ability to form (multiple) hydrogen bonds. The termination reactions R3, R4a, R6a, and R7 only form one more functional group of the respective HOM. Except of the functional group added by the termination reaction, distributions of functional groups are the same for all monomer termination products originating from the same HOM peroxy radical. This also includes HOM-ON. Hence, no substantial differences should be expected for the vapor pressures of all monomer termination products produced originating from the same HOM peroxy radical.

Considering the molecular mass instead of O-atoms of the HOM moiety, HOM-ON have higher vapor pressure compared to HOM-PP (Peräkylä et al., 2020), despite the heavier termination group -$ONO_2$ compared to -OOH, =O, or -OH. Note, that the number of functional groups cannot be inferred one to one from the number of O-atoms. HOM-peroxy radicals can be formed via autoxidation of peroxy radicals or via H-shifts and $O_2$ addition in alkoxy radicals and thus there may be different numbers of O-atoms per functional groups. In addition, peroxy radicals may have the same molar weights but may have different molecular structures. With this limitation in mind, we will exploit the relationship between $\gamma_{eff}$ and O atoms as proxy for the number of functional groups in the next step of interpretation.

Starting from a given HOM-peroxy radical, the number of functional groups in the termination products is the same, independent if a HOM-PP or a HOM-ON is being formed. However, the masses of HOM-PP and HOM-ON differ. If a HOM-ON is formed, it contains 1 N and 1 or 2 O-atoms more than the parent HOM peroxy radical (depending on NO or $NO_2$ being added). If a HOM-PP is formed in a reaction of the same HOM peroxy radical with another peroxy radical, $HO_2$ or $RO_2\cdot$, the number of O-atoms stays constant in case of hydroperoxide formation or decreases by 1 in case of ketone or alcohol formation. This means that the formation of HOM-ON generally increases the molecular mass compared to a HOM-

PP (as was considered in Figure 6). Since the $\gamma_{eff}$ are similar for all monomer HOM originating from the same HOM peroxy radical, i.e. HOM-PP and HOM-ON have similar vapor pressures, a gain of SOA mass could be expected if HOM-ON are produced instead of HOM-PP ($\approx$10% gain per HOM-ON for NO and at a molecular mass of 300 Da). However, whether this potential mass gain can be realized at all in a long term net increase of SOA mass in $NO_X$ containing systems will depend on the fate of the ON in the particle phase.

## 4.3 Comparison of OrgNO$_3$ in HOM to OrgNO$_3$ in particles

Comparing the data shown in Figure 8, it is obvious that the mass fraction of OrgNO$_3$ in gas-phase HOM is 3 to 4 times higher than that found in the particulate phase. As all considered HOM (molecular mass > 230 Da) have a high efficiency for SOA formation, a fraction of about 10 % OrgNO$_3$ would be expected in the particle phase. Zhao et al. (2018) performed experiments of α-pinene photo-oxidation in the presence of 20 ppb $NO_X$ at a lower relative humidity of $\approx$30% in the SAPHIR chamber in Jülich. Of course there are differences between steady state experiments in JPAC and time dependent experiments in the batch reactor SAPHIR. However, applying the same instrumentation and using the same evaluation schemes as here, Zhao et al. (2018) find a mass fraction of 11% for OrgNO$_3$ in the particulate phase, i.e. mass closure as expected if all HOM with more than 6 O-atoms contribute to SOA formation. This is more than the 2-3% OrgNO$_3$ that we realized in the particulate phase. A difference between the experiments here in JPAC and Zhao et al. (2018) in SAPHIR was the relative humidity which was about 30 % by Zhao et al. (2018) and 63% in the study here. We suggest that the RH may be the key to bring this study and the results by Zhao et al. (2018) in agreement, although we cannot provide further experimental proof. However, there are several studies implying that ON undergo hydrolysis in the condensed phase (Bean and Hildebrandt Ruiz, 2016; Boyd et al., 2015; Browne et al., 2013; Day et al., 2010; Fisher et al., 2016; Jacobs et al., 2014; Lee et al., 2016b; Liu et al., 2012; Rindelaub et al., 2016, 2015). Thus, hydrolysis can be suspected as a mechanism explaining both, the lower fraction of OrgNO$_3$ in the particle phase compared to that stored in HOM organic nitrates and the lower amount or OrgNO$_3$ found in this study compared to that found by Zhao et al. (2018).

In addition, hydrolysis of OrgNO$_3$ could contribute to SOA mass suppression. We estimated the effect of ON hydrolysis on the remaining SOA mass under the assumption that all HOM-ON with $\gamma_{eff}$ > 0.5 had condensed on SOA. By hydrolysis of organic nitrates HNO$_3$ is formed and the nitrate functionality at the organic rest is replaced by an OH group (Hu et al., 2011). HNO$_3$ is too volatile to stay in the particle phase (e.g., Browne et al., 2013; Romer et al., 2016) and we found only negligibly small amounts of inorganic nitrate in particles despite the high HNO$_3$ production in the gas phase (section 3.1). As a consequence, on average 1/5 of the mass of the HOM-ON that originally condensed on particles might re-evaporate and indeed reduce the amount of condensed mass. The question is what happens to the remaining organic moiety. As long as hydrolysis does not lead to fragmentation of the organic rest, hydrolysis just replaces the nitrate group by the protic OH group. The number of functional groups remains the same and no strong changes in the vapor pressures are expected for the organic rest. As a consequence the organic rest of the former organic nitrate would stay in the particle phase. If so,

hydrolysis and re-evaporation of $HNO_3$ should not lead to a mass loss high enough to explain the often observed suppressing effect of $NO_X$ on SOA mass formation in laboratory studies. The mass loss per evaporated $HNO_3$ is 63 Da. The water molecule driving the hydrolysis can be from gas-phase or particulate phase and exchange of water between the phases is fast. The de facto mass loss therefore should be 45 Da per evaporating $HNO_3$. A part of this loss is "compensated"; depending on

the HOM-ON being formed by NO or $NO_2$ and, depending on the HOM-PP not produced instead of the organic nitrate, this mass gain is between 29 Da and 63 Da per formed HOM-ON. Thus, the net effect of mass gain by the formation of a HOM-ON instead of a HOM-PP in the gas phase and the possible mass loss due to evaporation of $HNO_3$ from the particle phase should be very low or negligible.

## 4.4 Suppression of accretion products and SOA yield

In the previous section we have shown that the formation of organic nitrates instead of other HOM monomer termination products cannot explain the suppressing effect of $NO_X$ even if we consider hydrolysis and loss of $HNO_3$. Also increasing fragmentation via alkoxy radicals from reaction R6b seemed to play only a minor role (see section 3.3, Figure 3). Reasons are that fragments formed in α-scission can still be highly functionalized, thus are simply HOM with less C atoms. In addition bond scission in alkoxy radicals of α- and β-pinene can lead to ring opening retaining the carbon number.

Moreover, isomerization of alkoxy radicals is another pathway (unimolecular or bimolecular) that leads eventually to peroxy radicals with same number of carbon atoms that could undergo further autoxidation and/or terminate to highly functionalized HOM (Vereecken et al. 2009, 2010).

The most probable explanation for the $NO_X$ induced suppression of SOA formation in laboratory studies is the suppression of HOM-ACC formation (compare Rissanen, 2018). At low $NO_X$ conditions, the mass fraction of HOM-ACC is similar to

those of monomers and it dropped to less than 30% at high [$NO_X$] (Figure 3). Hence, besides reduction of [OH] at high [$NO_X$], suppression of HOM-ACC formation by $NO_X$ might lead to a strong suppression of SOA formation in laboratory studies.

The precursors of the respective HOM-ACC are the key to understand how the suppression of HOM-ACC formation could lead to a suppression of SOA formation. As we showed in Figure 3 HOM-ACC decreased with $NO_X$, while HOM monomers

remained about constant. We therefore analyze how a reduction of SOA could be realized when HOM monomers (here HOM-ON) are formed instead of HOM-ACC. According to reaction R5, HOM accretion products are produced from two peroxy radicals (Berndt et al., 2018a, 2018b). The other product of this reaction is molecular oxygen and thus the molecular mass of the HOM accretion product is lower by 32 Da than the sum of the molecular masses of both monomer peroxy radicals. If two HOM-ON are formed from two HOM-$RO_2$ by reactions R6a and R7 instead of one HOM-ACC, each of them

gains molecular mass due to the addition of NO (30 Da) or $NO_2$ (46 Da). Comparing the molecular mass of two HOM-ON to that of the HOM-ACC formed from the same HOM-$RO_2$, there may be even a gain of 92 Da to 124 Da when two HOM-ON are formed on cost of one HOM-ACC. This gain considers the addition of two NO or two $NO_2$ in the formation of HOM-ON

and the loss of $O_2$ in HOM-ACC formation. Hence, if the respective HOM-ACC is formed by HOM-RO$_2$ radicals that anyhow would form low volatility HOM-ON, the suppression of HOM-ACC should actually lead to an increase of total condensable mass.

The situation is different if we assume that classical ("non SOA forming") RO$_2$· with lower O:C ratio were involved in HOM-ACC formation besides HOM-RO$_2$. If the RO$_2$ is not terminated by a HOM-RO$_2$ to HOM-ACC, the volatility of its classical termination products may be too high to allow for effective condensation and contribution to SOA formation. In such cases, HOM-ACC formation would lead to a gain of condensable mass by scavenging the "non SOA forming" peroxy radical. In turn, suppression of HOM-ACC formation would indeed lead to a net loss of condensable mass. This effect is de facto the same as accretion product suppression by isoprene peroxy radicals described by McFiggans et al. (2019).

The loss could be even stronger when two intermediate level oxidized (functionalized) peroxy radicals are involved in the accretion product formation. If both form volatile termination products otherwise, the whole accretion product accounts for loss. Involvement of "non-SOA forming" RO$_2$· in HOM-ACC formation can be verified by looking at average O:C ratios derived from high resolution peak identification. For α-pinene at the background level of NO$_X$ the average O:C is 0.97 for the monomers and 0.68 for the accretion products. The lower O:C ratios of HOM-ACC indicate a substantial contribution of RO$_2$· with smaller numbers of O atoms.

Since HOM-ACC can be formed from many permutations of HOM-RO$_2$ and "non-SOA forming" RO$_2$·, clear identification of the respective precursors is not possible. Referring to rate coefficients reported by Berndt et al. (2018b), which decrease with the degree of functionalization by two orders of magnitude, we propose that there may be three types of pathways to accretion products: HOM-RO$_2$· + HOM-RO$_2$·, HOM-RO$_2$· + RO$_2$·, and RO$_2$· + RO$_2$·. Formation of accretion products by reaction HOM-RO$_2$· + RO$_2$· were observed for cyclopentene by Mentel et al. (2015). For illustration we simply assume that accretion product formation involves a pre-stabilized adduct (in analogy to the Lindemann-Hinshelwood mechanism). Then HOM-RO$_2$· + HOM-RO$_2$· would form a relatively long lived and relatively stable adduct because of the high functionalization of the reactants (with protic functional groups). Such adduct would live long enough to react to the accretion products e.g. as proposed by Valiev et al. (2019). HOM-RO$_2$·+RO$_2$· form weaker adducts with shorter lifetimes, but there are more collisions to form adducts as [RO$_2$·] are higher than [HOM-RO$_2$·]. RO$_2$· + RO$_2$· may still take place, driven by bare number of collisions. All involved RO$_2$· must have certain degree of functionalization (Berndt et al. 2018a, b). First generation RO$_2$· contain only 3 O-atoms, but are by far the most abundant. Reactions of first generation peroxy radicals could therefore still make a contribution to accretion products.

We conclude that suppression of HOM accretion product formation is a mechanism that leads to lower amounts of condensable mass because of involvement of "non SOA forming" RO$_2$· and therefore can explain the suppressing effect of NO$_X$ on SOA formation. Note that in the experiments here RO$_2$· dominated over HO$_2$·. This is often the case in laboratory studies with enhanced VOC and oxidant levels. Insofar, suppression of accretion products may well explain the dependence of SOA formation on [NO$_X$] (and the variability) observed in laboratory studies. In the atmosphere, photochemical accretion

product formation at low [NO$_X$] can often be less important because termination reactions with HO$_2$ are more important for HOM formation than termination reactions with RO$_2\cdot$ (compare Berndt et al., 2018a for example of isoprene).

## 5 Summary and conclusion

We characterized the role of ON for SOA mass formation. One finding was that low functionalized ON do not contribute much to particle formation. Only HOM-ON with more than 6 O-atoms at the HOM moiety can efficiently contribute to SOA mass formation at least at mass loads as investigated here. Thereby HOM-ON with 6 to 7 O atoms showed partitioning with $\gamma_{eff}$ of about 0.5, i.e. about 50% staying in the particulate phase. Once the HOM-ON contained more than 8 O-atoms their loss on particles was collision limited and nearly 100% resided in the particulate phase. This supports expectations that HOM-ON with more than 8 O-atoms will have extreme low volatility. No significant and systematic differences in $\gamma_{eff}$ were found between HOM-ON and HOM-PP when they have the same number of O-atoms in the moiety, i.e. when they arise from the same HOM-RO$_2$. Hence, different volatility of HOM-ON and HOM-PP from different termination reactions can be discarded as reason for the suppressing effect of NO$_X$ on SOA mass formation. Hydrolysis of HOM-ON in the particle phase and re-evaporation of HNO$_3$ seems also insufficient to explain the suppressing impacts of NO$_X$ on SOA mass formation. Re-evaporation of HNO$_3$ more or less just compensates the mass gain due to the formation of a HOM-ON instead of a HOM-PP. Thus we conclude that formation of HOM-ON instead of HOM-PP (i.e. hydroperoxides, -alcohols, -ketones, -carboxylic or -percarboxylic acids) cannot be the main reason for the often observed suppressing effect of NO$_X$ on SOA formation in photochemical systems. Since a suppressing effect of NO$_X$ on SOA mass formation is well documented in the literature (Presto et al., 2005; Kroll et al., 2006; Ng et al., 2007; Eddingsaas et al., 2012; Sarrafzadeh et al., 2016; Stirnweis et al., 2017), there must be other mechanisms causing this suppression. One effect is the lowering of the OH level by NO$_X$ (e.g. Sarrafzadeh et al., 2016; Lee et al. 2020). If OH is kept constant, we observed strong suppression of HOM-ACC with increasing [NO$_X$]. Formation of HOM-ON via fast R6a, R7 in competition to R4a is probably the reason of the observed phenomenon, especially if R6a and R7 prevent that less functionalized RO$_2\cdot$ are trapped in low volatility accretion products. Without forming accretion product, there is no chance for them to participate in SOA mass formation because of the high volatility of their other termination products. Keeping off less functionalized HOM-RO$_2$ from forming accretion products leads to loss of one molecular mass unit of less functionalized RO$_2\cdot$ or even the whole accretion product if it was formed by two intermediate level oxidized peroxy radical. This effect may be less expressed in the atmosphere as RO$_2\cdot$ - RO$_2\cdot$ interactions in low NO$_X$ cases are less important than in our laboratory study.

There is another contribution left for an explanation of SOA mass suppression by NO$_X$: the decomposition of alkoxy radicals that are formed in reaction R6b. We showed in Figure 3 that fragmentation of alkoxy radicals led to C$_{<10}$ compounds, that are still HOM that contribute to SOA. At the current stage the overall impact of alkoxy radicals on HOM and SOA formation is difficult to address and needs closer study (in preparation).

Note, that we considered the photochemistry of $NO_X$ to SOA contribution for two major MT, α-pinene and β-pinene. We find that that SOA yields are fairly independent of $[NO_X]$, but drop significantly at the highest $NO_X$ levels. Model studies show that increase of $NO_X$ emissions may also lead to more SOA, when $NO_3$ is the oxidant (e.g. Pye et al. 2015) or when isoprene is involved (Marais et al. 2016). In the latter case NO directs the gas phase mechanism toward isoprene products with reactive uptake, while for compounds like α-pinene and β-pinene, investigated here, condensation is more important for SOA formation and thus vapor pressures controls SOA yields.

**Data availability**

All data given in figures can be displayed in tables or in digital form. This includes the data given in the Supplement where we describe methodological issues including calibrations, peak separation in CIMS, and peak lists. Please send all requests for data to t.mentel@fz-juelich.de.

**Author contributions**

TFM, JW, EK, IP, AW, and AKS designed the experiments. Instrument deployment and operation were carried out by IP, SS, MS, PS, SA, EK, FR, MS, RT, JW, CW, and DZ. Data analysis was done by IP, SK, SS, FR, PS, and RT. IP, SK, SS, JW and TFM interpreted the compiled data set. IP, SK, JW, TFM, and AKS wrote the paper. All co-authors discussed the results and commented on the paper.

**Competing interests**

Sebastian Schmitt is working for TSI GmbH. The authors declare that they have no conflict of interest.

**Acknowledgements**

This work was supported by the European Commission's 7th Framework Program under grant agreement no. 287382 (Marie Curie Training Network PIMMS). We would like to thank the anonymous reviewers for their helpful comments, especially we acknowledge anonymous reviewer 2 who helped a lot to improve the manuscript in the second round.

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

**Tables**

**Table 1: Overview of α-pinene and β-pinene experiments**

| Experiment Description | $[VOC]_0{}^a$ [ppb] | $[NO_X]_0{}^a$ & $([NO_X]_{SS}{}^b)$ [ppb] | $[O_3]_{SS}{}^b$ [ppb] | $[OH]_{SS}{}^b$ $[10^7 cm^{-3}]$ |
|---|---|---|---|---|
| 1. Gas-phase yield of ON and gas-phase OrgNO₃ (Section 3.1) | β-pinene 39→0 m-xylene 3.7 | 50 (20→30) | 19→30 | 2.3±20% |
| 2. Formation of HOM-ON (Section 3.3) | α-pinene 16.5 | 0.3 / 7.5 / 15.3$^c$ / 26.7 / 39.7 / 45.5 (0.3 / 1.8 / 3.7$^c$ / 5.7 / 8.7 / 10.4) / 52.9 / 59.1 / 83.3 / 137.8 (/ 12.4 / 15.8 / 26.8 / 72.2) | 62 -152 | 4.5 -7.5 |
| | β-pinene 37 | 3.9 / 53.8 / 113.6 / 194 (1.2 / 16.5 / 37.0 / 77.) | Not determined | Not determined |
| 3. Effective uptake coefficients$^d$ (Section 3.4) | α-pinene 12.5 | 0.3 (0.3) | 29 | 9.2±20% |
| | β-pinene 37 | 30 (4) | 49 | 8.8±20% |
| 4. OrgNO₃ in SOA (Section 3.5) | α-pinene 46 | 0.3 / 32.0 / 51.0 / 60.0 (0.3 / 10.4 / 17.5 / 19.5) | 37 - 62 | 4.7- 7.7 |
| | β-pinene 38 | 0.3 / 6.7 / 13.4 / 32.9 / 54.8 / 103 (0.3 / 5.1 / 9.5 / 21.7 / 35.5 / 45.7) | 44 – 53 | 0.9 - 3.7 |

[a] subscript ₀ refers to mixing ratio in the inflow
[b] subscript SS refers to mixing ratio at steady state
[c] average of two experiments at $[NO_X]_0$ of 15 and 15.5 ppb ($[NO_X]_{SS}$ of 3.6 and 3.75 ppb)
[d] in presence of ammonium sulfate seed aerosols

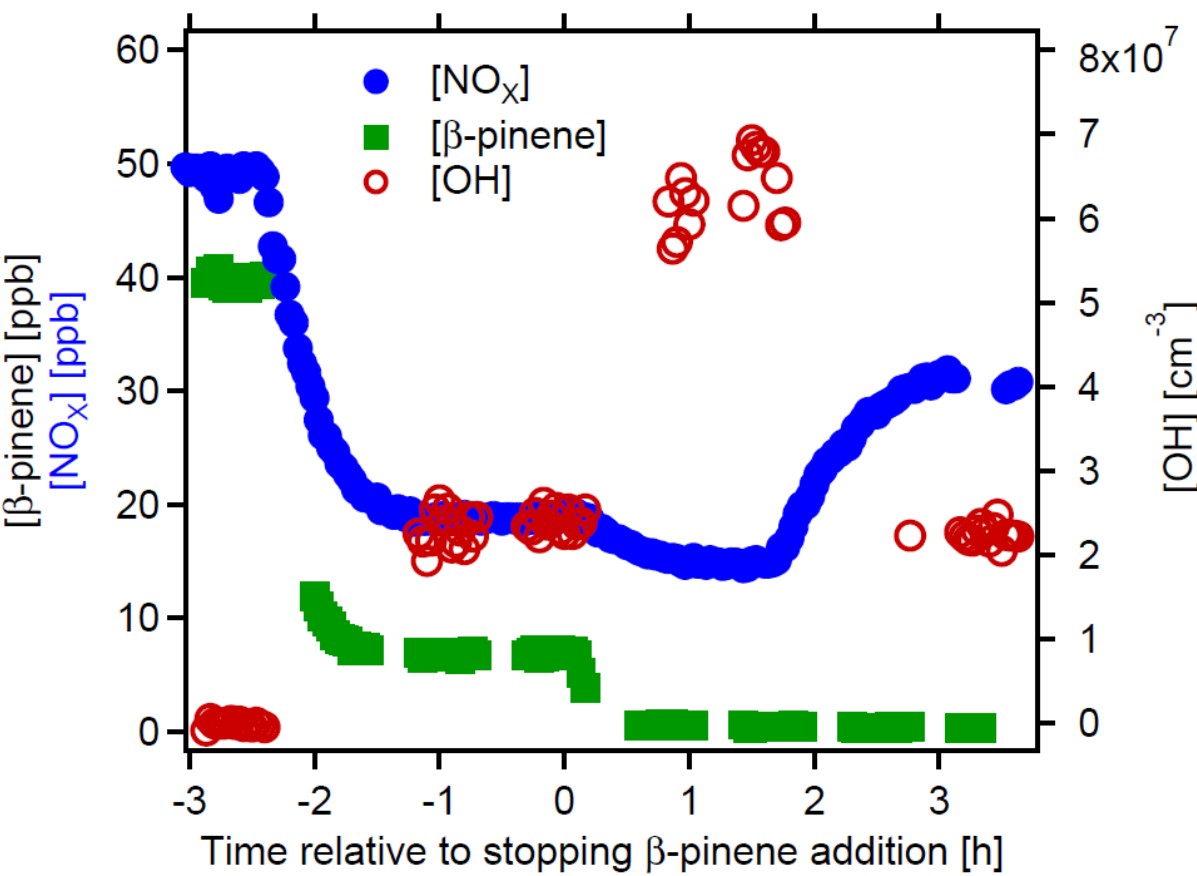

Figure 1: Time series of [β-pinene] (green squares, left scale), [NO$_X$] (blue circles, left scale) and [OH] (open brown circles, right scale). The experiment served to estimate the sum of organic nitrates (ON) formed in a mix of NO$_X$ and β-pinene. M-xylene ([m-xylene]$_0$ ~ 3.7 ppb) was added to the chamber as tracer for OH. At time t = -2.4 h OH formation was induced by O$_3$ photolysis. At time t = 0 h, β-pinene addition was stopped and at time t = 1.7 h J(O$^1$D) was reduced to obtain the same [OH] as in presence of β-pinene at time -1 h.

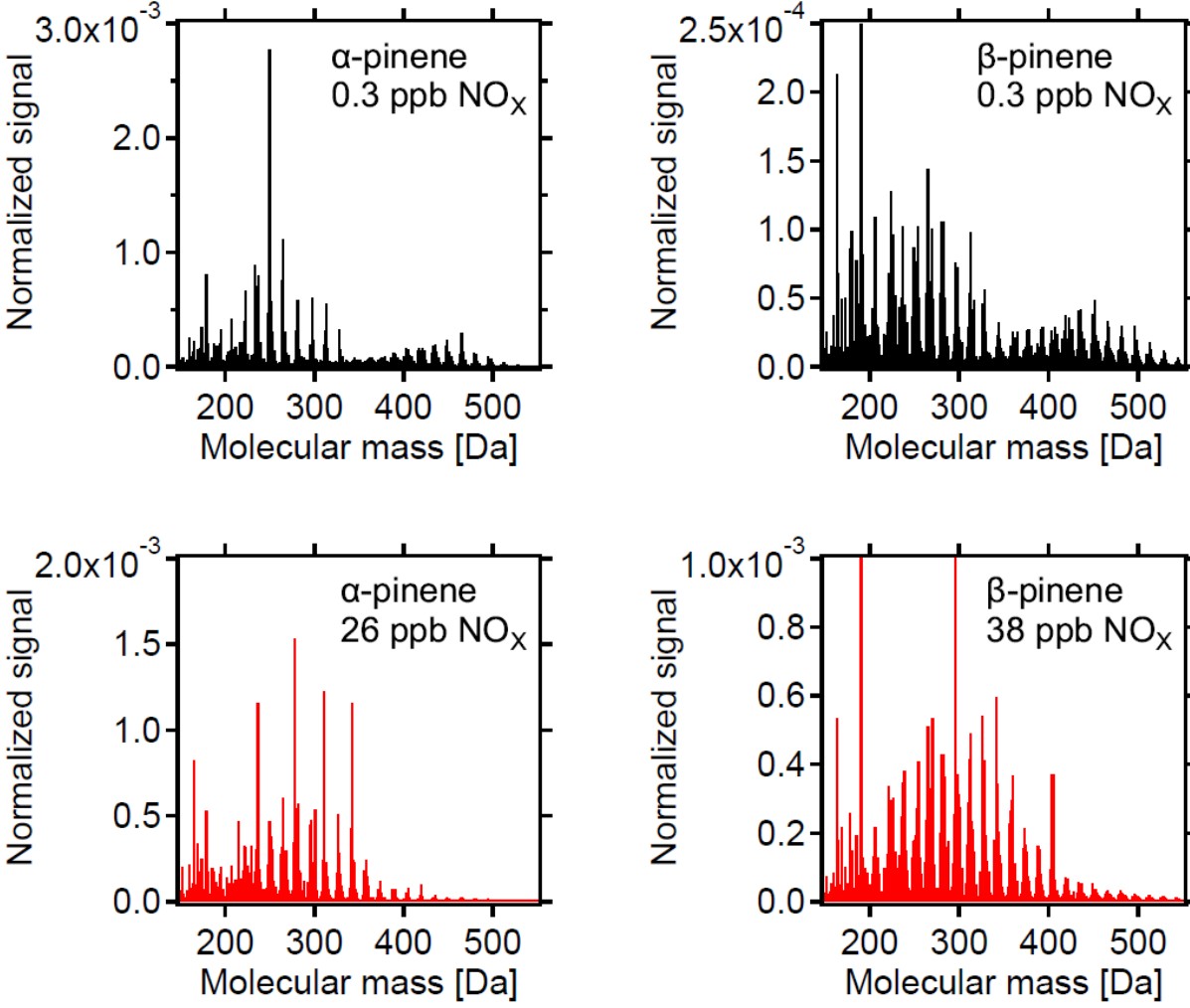

**Figure 2. HOM spectra from photo-oxidation of α-pinene (left panels) and β-pinene (right panels) without NO$_X$ addition (upper panels) and with NO$_X$ addition (lower panels). NO$_X$ concentrations in the α-pinene and β-pinene experiment were 26 ppb and 38 ppb, respectively. Background NO$_X$ was 0.3 ppb. The signals were normalized to the sum over all detected ions. For the α-pinene example, in the low NO$_X$ case HOM monomers contribute ≈0.4 µg m$^{-3}$ and HOM-ACC ≈0.3 µg m$^{-3}$, whereas at 26 ppb NO$_X$ HOM monomers contribute ≈0.4 µg m$^{-3}$ and HOM-ACC less than 0.1 µg/m$^{-3}$ (compare Figure 3).**

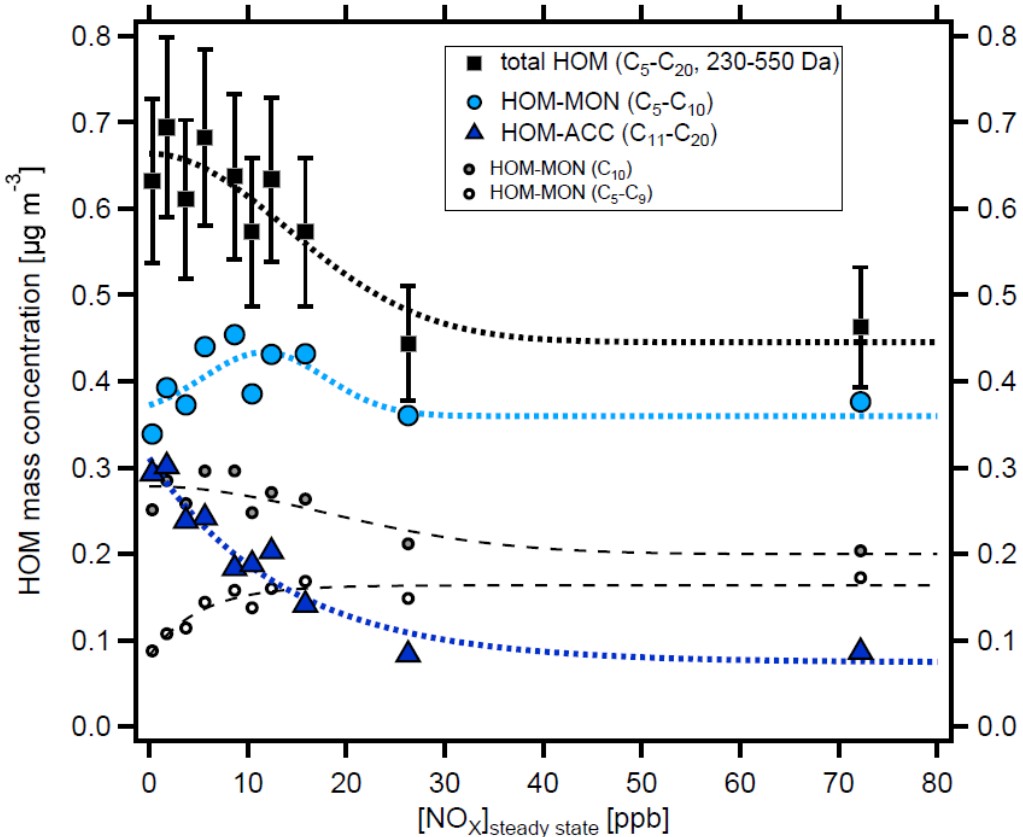

**Figure 3. Mass concentration of HOM products in dependence on $[NO_X]_{SS}$ in α-pinene photo-oxidation experiments. $C_5$-$C_{20}$ compounds with molecular masses 230-550 Da were added up for total HOM (black squares) and divided into HOM monomers (light blue circles) and HOM accretion products (blue triangles). The analysis is based on the assigned peaks (>90% of the total signal) and the sensitivity of $3.7 \times 10^{10}$ molecules $cm^{-3}$ $nc^{-1}$ (suppl. section 1.2). Dashed and dotted lines save to guide the eye and have no further meaning. Concentrations were corrected as described in supplement section S1.2. Turnover ranged from $8.7 \times 10^7$ $cm^{-3} s^{-1}$ and $1.04 \times 10^8$ $cm^{-3} s^{-1}$ leading to correction factors in a range of 1.1 - 0.8. The correction factors were close to one thus did not add much uncertainty. Observed particle surface ranged from $\sim 10^{-6}$ $m^2 m^{-3}$ to $6 \times 10^{-5}$ $m^2$ $m^{-3}$ resulting in correction factors between 1.0 and 1.45 with the highest correction factors at lower $[NO_X]_{SS}$ where new particle formation could not be suppressed.**

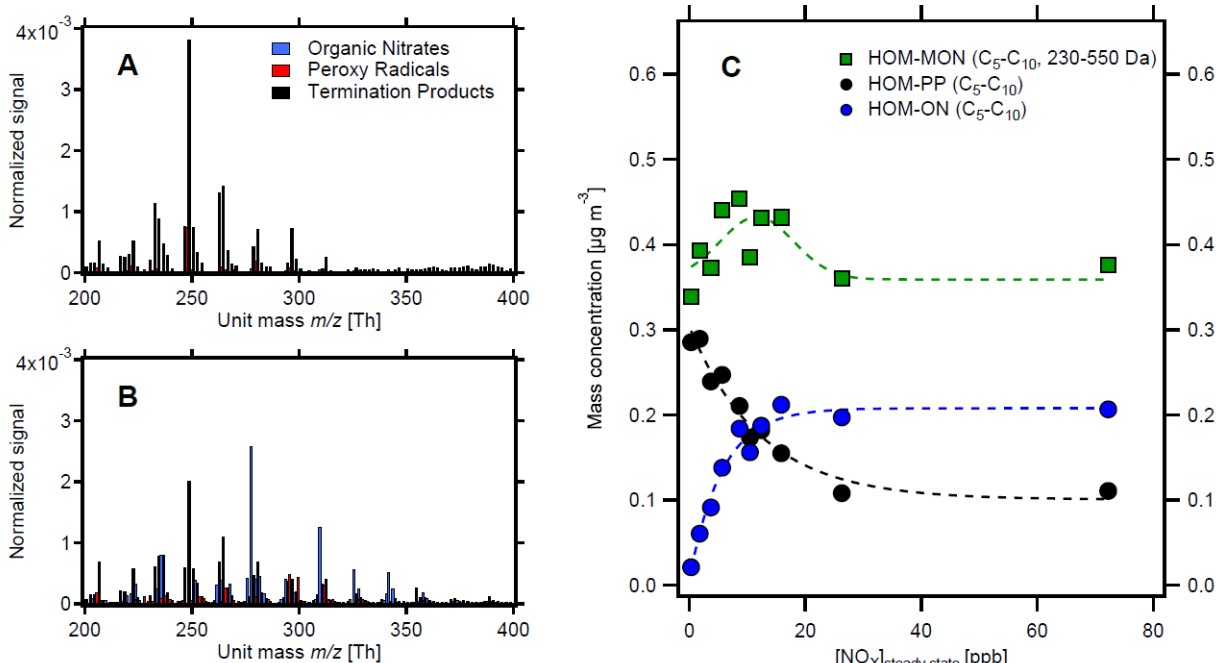

Figure 4: HOM pattern from α-pinene photo-oxidation at two NO$_X$ levels in the monomer range. Panel A: low NO$_X$ conditions ([α-pinene]$_{SS}$ = 1.7 ppb, [NOX]$_{SS}$ = 0.3 ppb), Panel B: high NO$_X$ conditions ([α-pinene]$_{SS}$ = 1.0 ppb, [NOX]$_{SS}$ = 8.7 ppb). Black bars: HOM-PP termination products of reactions R3 and R4a. Blue bars: HOM-ON (organic nitrates). Red bars = HOM-RO$_2$ (peroxy radicals). The signals were normalized to the sum over all detected ions. Panel C: Mass concentrations of HOM monomers (green) in the molecular mass range 230-550 Da. HOM-ON (blue) are increasing with increasing [NO$_X$]$_{SS}$, HOM-PP (black) are decreasing, while the sum of all HOM-monomers remains about the same. At about 10 ppb [NO$_X$]$_{SS}$ HOM-ON make up half of the HOM monomers and at 26 ppb [NO$_X$]$_{SS}$ they make up about 50% of the total HOM (shown in Figure 3). Dashed lines serve only to guide the eye.

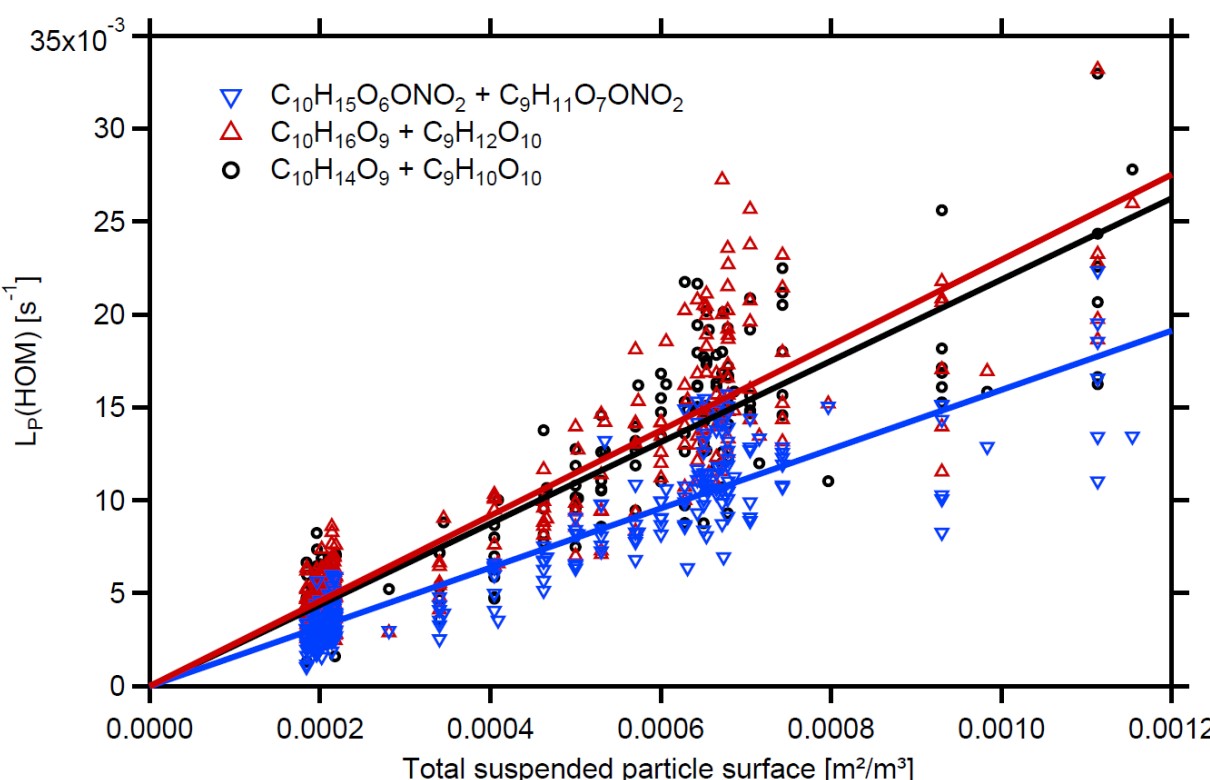

5 **Figure 5:** Plot of $L_P(HOM)$ calculated by Eq. (6) versus particle surface area, $S_P$, for the examples of HOM-ON with a molecular mass of = 293 Da ($C_{10}H_{15}O_6ONO_2$ & $C_9H_{11}O_7ONO_2$), HOM-PP with a molecular mass of 280 Da ($C_{10}H_{16}O_9$ and $C_9H_{12}O_{10}$), and HOM-PP with molecular mass of 278 ($C_{10}H_{14}O_9$ and $C_9H_{10}O_{10}$). HOM from β-pinene photo-oxidation ([β-pinene]$_{SS}$ ~ 10 ppb, [NOx]$_{SS}$ ~ 4 ppb). Dividing the slopes by the respective $\tilde{v}/4$ led to $f_{FS} \times \gamma_{eff}$ ~ 0.5 for the example HOM-ON and ~ 0.6 in the latter cases. The main uncertainty arises from the scatter of $L_P(HOM)$ individual unit mass resolution data. Statistical errors of $f_{FS} \times \gamma_{eff}$

10 were about ± 5%.

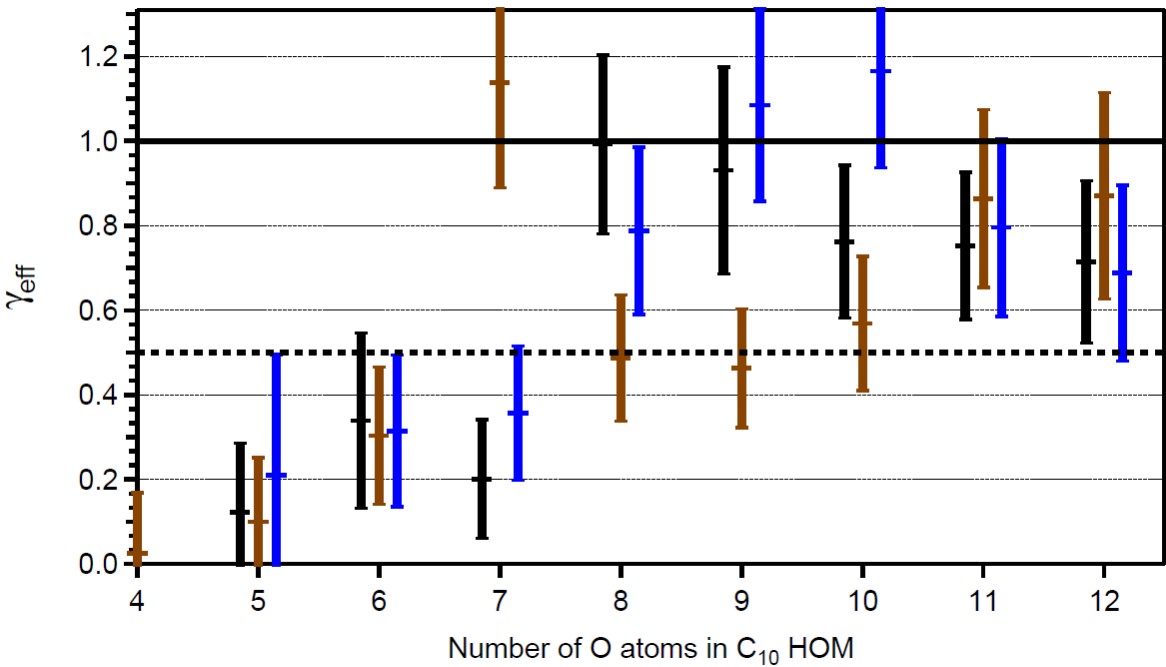

**Figure 6: Effective uptake coefficients $\gamma_{eff}$ for HOM-PP ($C_{10}H_{14}O_X$ black bars, $C_{10}H_{16}O_X$, brown bars) and HOM-ON ($C_{10}H_{15}O_XNO_2$, blue bars) in dependence of the number of O atoms in the respective HOM. HOM with different numbers of C, H, and O atoms, e.g. $C_{10}H_yO_x$ and $C_9H_{y-4}O_{x+1}$ HOM-PP, are treated together and the number of O-atoms is given for the $C_{10}$-HOM-PP. The second component, $C_9$-HOM-PP, has one O atom more. Data were taken from β-pinene photo-oxidation experiment with [β-pinene]$_{SS}$ ~10 ppb, [NO$_X$]$_{SS}$ ~ 4 ppb. The signal intensity for the $C_{10}H_{14}O_4$ and HOM-ON with 4 O-atoms was too low to allow reliable determination of $\gamma_{eff}$ and the respective data is left out. Uncertainties in $\gamma_{eff}$ arise from the determination procedure as shown in Figure 5. The black line indicates $\gamma_{eff}$ = 1 and 0.5. An average Fuchs-Sutugin correction factor of 0.70 ($d_p$ =175nm) was applied to calculate $\gamma_{eff}$.**

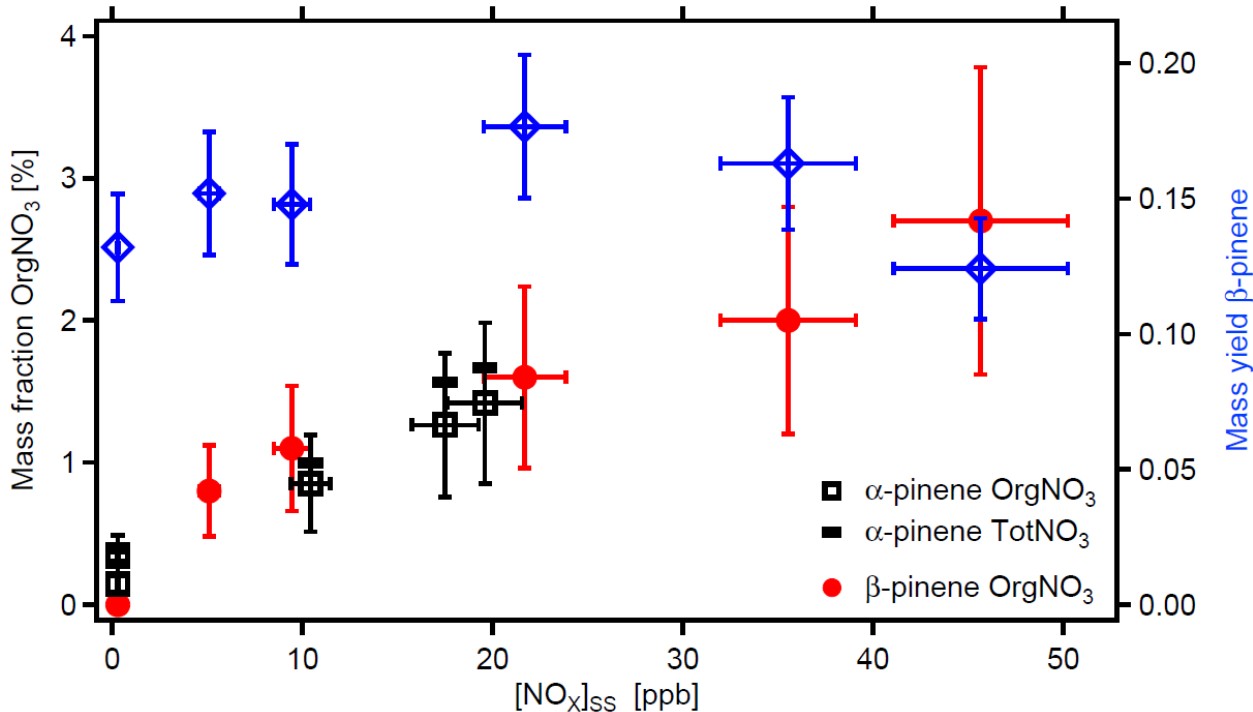

**Figure 7: Mass fraction of organic bound nitrate (OrgNO$_3$) in SOA in dependence of [NO$_X$]$_{SS}$ (left y-axis). Black squares and red circles show data measured from α-pinene and β-pinene, respectively. SOA mass yields during the respective experiment are shown at the example of β-pinene (blue diamonds). The SOA load ranged from 11 µg m$^{-3}$ to 23 µg m$^{-3}$ with an average of 16±5 µg m$^{-3}$. The data are corrected for wall losses of HOM. In absence of OH, [α-pinene]$_0$ was around 46 ppb, [β-pinene]$_0$ was around 37 ppb. NO$_X$ was added at different amounts with [NO$_X$]$_0$ up to 103 ppb. Due to losses in reactions with OH and formation of organic nitrates, [NO$_X$] decreased to the [NO$_X$]$_{SS}$ levels shown here. Uncertainties in NO$_X$ data are estimated to ± 10%, uncertainties in SOA masses to ± 10 %, and uncertainties in the content of OrgNO$_3$ are estimated to ± 40%. The black bars indicate the fraction of total nitrate (TotNO$_3$, left scale) for the example of α-pinene, which is dominated by organic nitrate.**

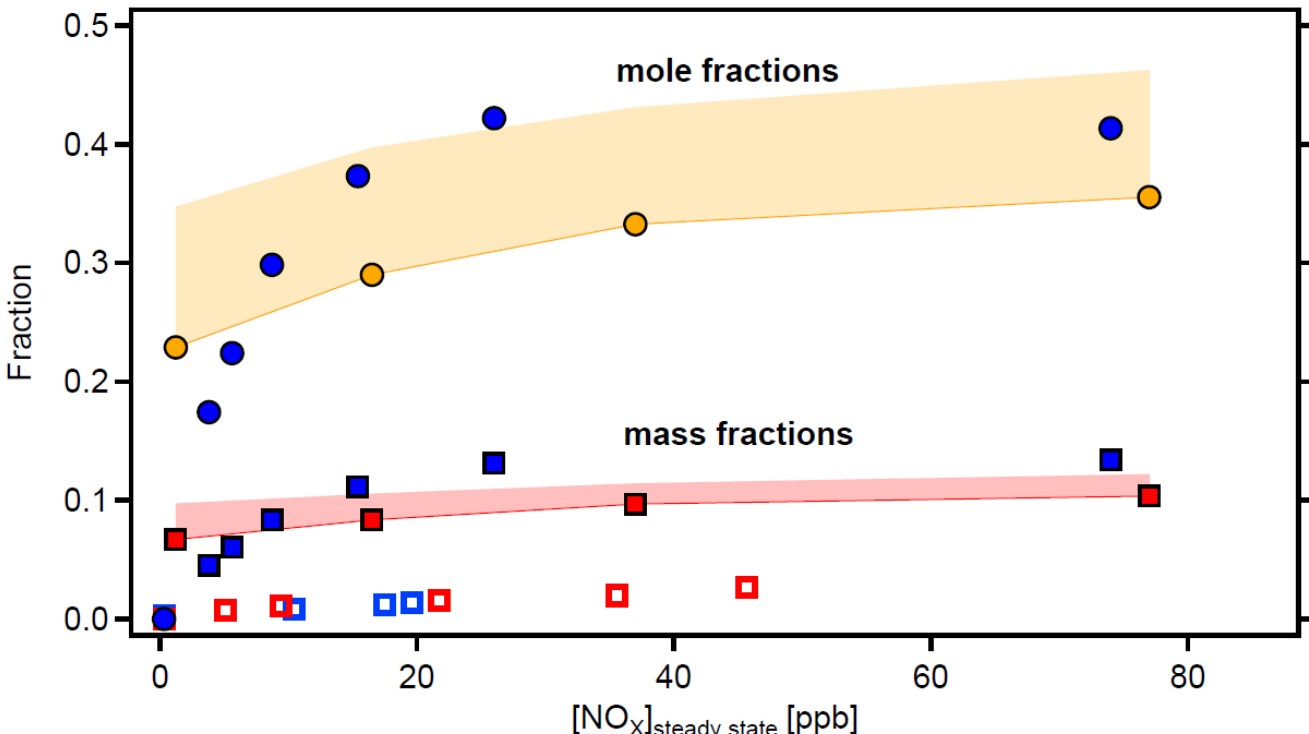

**Figure 8: Molecular fractions of organic bound nitrate (OrgNO₃, filled circles) and mass fractions of OrgNO₃ (squares) as a function of [NOₓ]ₛₛ. Data from α-pinene (blue symbols) and β-pinene (orange and red symbols and areas).** *Molecular fraction of* **OrgNO₃ and HOM-ON are the same by definition. The mass fraction of OrgNO₃ in the gas-phase HOM is significantly higher than in the particulate phase as determined by AMS (open blue and red squares).**

**The areas in orange and red give the potential error for β-pinene due to unresolved progressions and overlap of organic nitrates with peroxy radicals (as explained in text).**

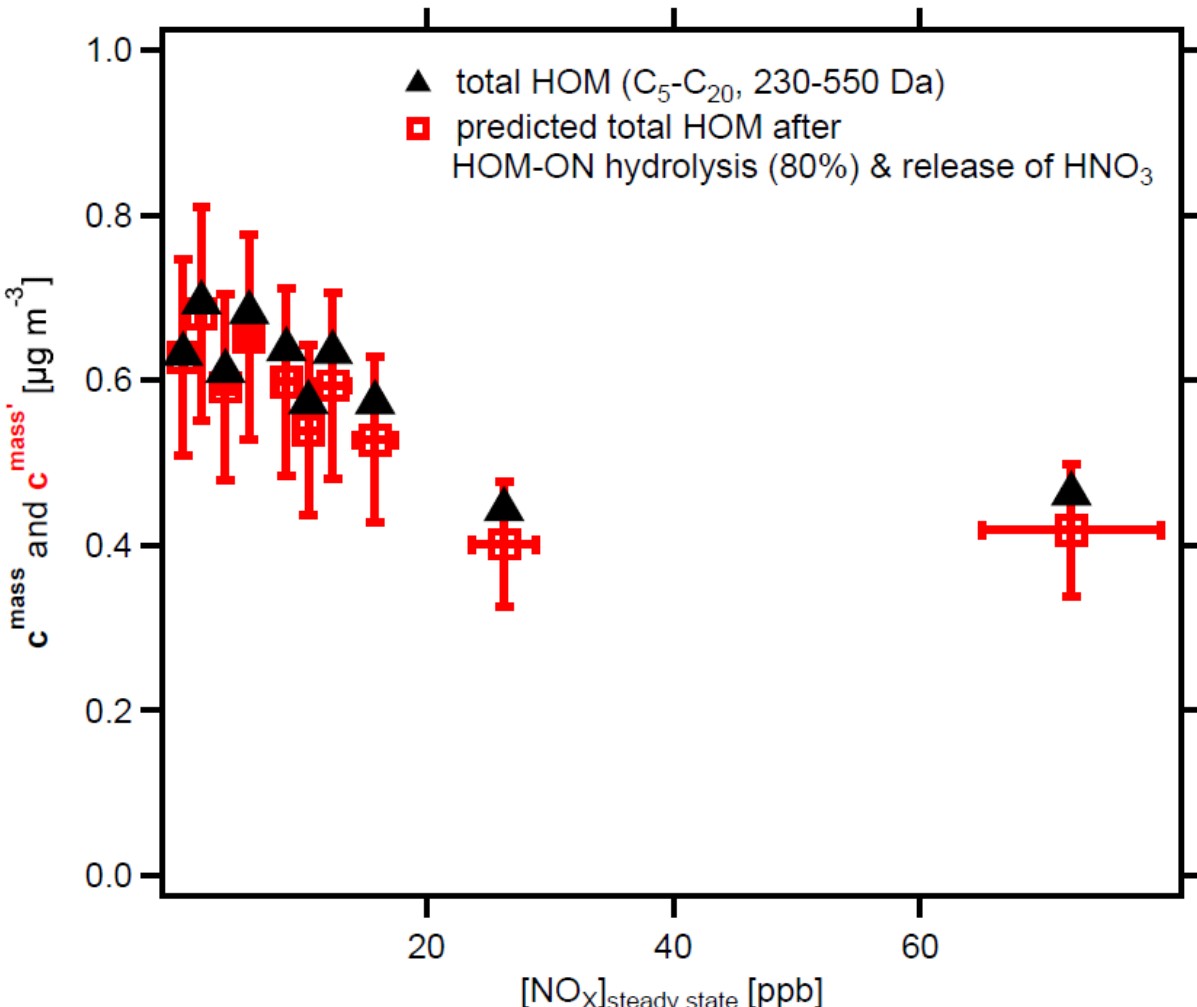

**Figure 9. Mass concentrations of total HOM ($C_5$-$C_{20}$) with molar masses between 230 to 550 Da. Black triangles show mass concentrations $c^{mass}$ as determined. Red squares show $c^{mass'}$ i.e. the resulting SOA mass after considering OrgNO$_3$ loss by hydrolysis and evaporation of HNO$_3$. [$\alpha$-pinene]$_{SS}$ = 0.9 to 2.2 ppb, [NO$_X$]$_0$ up to 125 ppb, [NO$_X$]$_{SS}$ = 0.3 to 74 ppb. The effect of hydrolysis of 80% of the organic bound nitrate has no substantial effect on the SOA mass. Analysis is based on assigned molecular formulas (>90% of the total signal) applying the sensitivity of $3.7\times10^{10}$ molecules cm$^{-3}$ nc$^{-1}$ (supplement section 1.2).**