# Peer review of "Impact of $NO_x$ on secondary organic aerosol (SOA) formation from $\alpha$ -pinene and $\beta$ -pinene photo-oxidation: the role of highly oxygenated organic nitrates"

_Atmospheric Chemistry and Physics, 2019_

## Referee Comment (RC1) · Anonymous Referee #1 · 10 Mar 2020

Pullinen et al. described an atmospheric simulation chamber work in the JPAC chamber on the photooxidation of $\alpha$-pinene and $\beta$-pinene at low and high NOx conditions at quite high humidity (relative humidity~63%). The main focus of the study is to investigate mechanism that NOX might suppress SOA formation in the respective experiments. The authors distinguished highly oxygenated multifunctional molecules (HOM), that can contribute to SOA yields, at low NOX levels as HOM-PP products (such as ketones, alcohols or hydroperoxides) and HOM accretion product with C>10 and C<20 (HOM-ACC). They attributed the NOX effect on suppression of the SOA mass to the

significant decrease of HOM-ACC as NOx increasing. When NOX was added to the reaction system HOM-PP was also decreased but HOM-organic nitrates (HOM-ON) concentrations increased on the cost of HOM-PP. and they find there was no systematic difference in effective uptake coefficients $\gamma$eff for HOM-ON at high NOx levels and HOM-PP at low NOx levels when they have the same number of O-atoms in the moiety. At the presence of ammonium sulfate seed particles, $\gamma$eff of HOM with more than 6 O-atoms determined to be > 0.5 in average and for HOM containing more than 8 O-atoms, $\gamma$eff $\sim$ 1. The finding makes a nice contribution to the literature in this area those have to simplify to estimate the uptake coefficients for particle organic nitrates (e.g. (Fisher et al., 2016; Marais et al., 2016)). I think it is an important and informative piece of work, providing experimental information to help current understanding of SOA formation under relevant atmospheric conditions. The subject of the paper is therefore directly within the ACP remit, and the manuscript is well written and the technical aspects and interpretations are reasonable. I recommend it be published in ACP after the following minor comments are addressed.

General points.

1. The paper concluded with increasing NOX HOM-ACC strongly decreases and consequently suppress SOA formation. While the experiments and analysis appear robust and in agreement with some literature, it is important to point out that some other literature such as Pye et al., (2015) and Marais et al. (2016), with specific representation of particulate organic nitrate predict the reduction in NOx emissions causes a considerable reduction in organic aerosol. Could authors comment on this discrepancy? In this regard, in Figure 2 authors showed HOM spectra with and without NOX addition. It might be worthwhile to mention total SOA or HOM mass for these two cases for easier comparison.

2. Page 11, Line 2 a) The authors estimated a molar yield of $\sim$36% for the ON formed from $\beta$ –pinene which is higher than the largest previously reported values (26 $\pm$ 7% Rindelaub et al., (2016), while for similar condition of this study (acidic seed aerosol

and RH~ 60%), they even estimated less (~6%). b) The authors mentioned at the Figure 9's caption "The effect of hydrolysis of 80% of the organic bound nitrate has no substantial effect on the SOA mass." However, Boyd et al. (2015) estimated particle-phase hydrolysis of organic nitrates compose 45–74% of the organic aerosol.

These discrepancies might be attributed to estimation of a slower aerosol hydrolysis in this study? and subsequently underestimation of importance of hydrolysis for explaining the SOA mass suppression with increasing NOX in the system?

3) Page 18, Line 11-24 The authors discussed higher humidity in their chamber and hydrolysis as the key for less estimation of OrgNO3 fraction in the particulate-phase than as determined by AMS and also finding (~11%) by Zhao et al. (2018). It is important to point out not only Zhao et al. but also many recent measurements (e.g., Romer et al., 2016) and modeling studies (e.g., Pye et al., 2015; Fisher et al., 2016; Zare et al., 2019) estimated a higher fraction of organic nitrates in the particle phase (~10%-20%). As they also considered a rather fast hydrolysis for organic nitrate aerosols it might be worthwhile to compare the result here to their results as well. However, it might be useful to mention Zare et al. show that at a more humid condition (similar to this study with higher RH) heterogeneous uptake to particle water tends to form less particulate organic nitrates against uptake to dry organic aerosols. Considering the impact of humidity at the aerosol formation together with the impact on the loss process of hydrolysis for particulate organic nitrates could help reconcile the discrepancy?

Minor comments

Page 3, for less confusion and similar to the other relevant papers, it is better to give a same reaction number for reactions with similar reactants, e.g. (R4) and (R4a) should be "(R4a)" and "(R4b)", and also for R5 and R7 should change as "(R5a)" and "(R5b)".

Page 6, Line 1-4, multiplication sign for reaction rates are missed.

Page 6, Line 2, References should be lined up in the proper sequence.

[Figure]

Page 13, Line 13, remove spare space before parentheses.

Page 15, Line 1-4, remove redundant parentheses.

Page 15, Line 21, remove extra "was"

Page 15, line 24, "estimated to be"

Page 32, Figure1, for better readability write axis label on the right-hand side from down to up, similar to the left-hand side of the figure.

Page 37, Line4, the brown bars look like more "orangeish" than brown in my eyes.

Page 40, Line 5-6 used different font.

References

Boyd, C. M., Sanchez, J., Xu, L., Eugene, A. J., Nah, T., Tuet, W. Y., Guzman, M. I. and Ng, N. L.: Secondary organic aerosol formation from the $\beta$-pinene+NO3 system: effect of humidity and peroxy radical fate, Atmos. Chem. Phys., 15(13), 7497–7522, doi:10.5194/acp-15-7497-2015, 2015.

Fisher, J. A., Jacob, D. J., Travis, K. R., Kim, P. S., Marais, E. A., Chan Miller, C., Yu, K., Zhu, L., Yantosca, R. M., Sulprizio, M. P., Mao, J., Wennberg, P. O., Crounse, J. D., Teng, A. P., Nguyen, T. B., St. Clair, J. M., Cohen, R. C., Romer, P., Nault, B. A., Wooldridge, P. J., Jimenez, J. L., Campuzano-Jost, P., Day, D. A., Hu, W., Shepson, P. B., Xiong, F., Blake, D. R., Goldstein, A. H., Misztal, P. K., Hanisco, T. F., Wolfe, G. M., Ryerson, T. B., Wisthaler, A. and Mikoviny, T.: Organic nitrate chemistry and its implications for nitrogen budgets in an isoprene- and monoterpene-rich atmosphere: constraints from aircraft (SEAC4RS) and ground-based (SOAS) observations in the Southeast US, Atmos. Chem. Phys., 16(9), 5969–5991, doi:10.5194/acp-16-5969-2016, 2016.

Marais, E. A., Jacob, D. J., Jimenez, J. L., Campuzano-Jost, P., Day, D. A., Hu, W., Krechmer, J., Zhu, L., Kim, P. S., Miller, C. C., Fisher, J. A., Travis, K., Yu, K., Hanisco,

T. F., Wolfe, G. M., Arkinson, H. L., Pye, H. O. T., Froyd, K. D., Liao, J. and McNeill, V. F.: Aqueous-phase mechanism for secondary organic aerosol formation from isoprene: application to the southeast United States and co-benefit of SO2 emission controls, Atmos. Chem. Phys., 16(3), 1603–1618, doi:10.5194/acp-16-1603-2016, 2016.

Pye, H. O. T., Luecken, D. J., Xu, L., Boyd, C. M., Ng, N. L., Baker, K. R., Ayres, B. R., Bash, J. O., Baumann, K., Carter, W. P. L., Edgerton, E., Fry, J. L., Hutzell, W. T., Schwede, D. B. and Shepson, P. B.: Modeling the Current and Future Roles of Particulate Organic Nitrates in the Southeastern United States, Environ. Sci. Technol., 49(24), 14195–14203, doi:10.1021/acs.est.5b03738, 2015.

Rindelaub, J. D., Borca, C. H., Hostetler, M. A., Slade, J. H., Lipton, M. A., Slipchenko, L. V. and Shepson, P. B.: The acid-catalyzed hydrolysis of an $\alpha$-pinene-derived organic nitrate: kinetics, products, reaction mechanisms, and atmospheric impact, Atmos. Chem. Phys., 16(23), 15425–15432, doi:10.5194/acp-16-15425-2016, 2016.

Romer, P. S., Duffey, K. C., Wooldridge, P. J., Allen, H. M., Ayres, B. R., Brown, S. S., Brune, W. H., Crounse, J. D., de Gouw, J., Draper, D. C., Feiner, P. A., Fry, J. L., Goldstein, A. H., Koss, A., Misztal, P. K., Nguyen, T. B., Olson, K., Teng, A. P., Wennberg, P. O., Wild, R. J., Zhang, L. and Cohen, R. C.: The lifetime of nitrogen oxides in an isoprene-dominated forest, Atmos. Chem. Phys., 16(12), 7623–7637, doi:10.5194/acp-16-7623-2016, 2016.

Zare, A., Fahey, K. M., Sarwar, G., Cohen, R. C. and Pye, H. O. T.: Vapor-Pressure Pathways Initiate but Hydrolysis Products Dominate the Aerosol Estimated from Organic Nitrates, ACS Earth Space Chem., 3(8), 1426–1437, doi:10.1021/acsearthspacechem.9b00067, 2019.

---

## Referee Comment (RC2) · Anonymous Referee #2 · 10 Mar 2020

The goal of this work is to explore the observed and documented suppression of SOA formation in the presence of NOx from previous studies. The authors hypothesized the observed decrease in SOA mass could be due to HOM termination reactions, in particular the reactions that form gas-phase accretion products or HOM organic nitrates. To test their hypothesis, they performed a series of chamber experiments at the Jülich Plant Atmosphere Chamber, a well characterized environment. They primarily used a Chemical Ionization Mass Spectrometer with NO3- ionization which has been repeatedly shown to be useful in measuring HOM, and studied the photooxidation of $\alpha$-

and $\beta$-pinene. They found that HOM accretion products were much more heavily suppressed than non-nitrate monomer products in the presence of NOx. They determine this loss of accretion products is a possible and even likely reason for SOA mass suppression in the presence of NOx. Section 4.4 discusses the suppression of accretion products but notably lacking is the possibility of precursor decomposition as a result of RO2 + NO (R7, page 3 line 24) forming an alkoxy which can rapidly decompose, resulting in smaller RO2 building blocks for accretion chemistry. This is brought up briefly on page 11 line 23, but is not worked into the discussion in section 4.4. It is however noted in the last sentence of the manuscript, only to say that it will be discussed in a further manuscript. This seems warranted to be discussed within this manuscript, and as-is I find it to be a major deficit of this work that should be discussed somewhere, potentially section 4.4. Additionally, the grammatical errors throughout impede understanding of the manuscript and need significant improvement before publication. All in all, this work is novel and of interest to the readers of ACP, so after addressing these major concerns and the other minor technical edits detailed below this manuscript should be suitable for publication.

Page 1, line 33-35: This could also be due to the higher vapor pressure of HOM-ON relative to similar non-nitrates as you reference on page 17 line 26-27. Consider rewording to make clear you're basing this statement off the results in this paper.

Page 2, line 28: "the absence of particles in the presence of NOx": consider rewording. The presence of NOx doesn't mean there's no particles around.

Page 5 line 28 – Page 6 line 7: It would be helpful to include the average or typical concentration of OH more clearly. It is state on Page 6 line 3 but it feels a bit buried and is a single value as opposed to the range stated for O3 and shown for OH in Fig. 1. Furthermore, the range of O3 stated on page 5 line 13 does not match that on page 6 line 2. I would suggest adding the O3 and OH concentrations to Table 1.

Page 8, line 10: please justify ignoring wall loss by providing a reference for particle

wall loss rates on similar chambers, or a loss rate estimate from your measurements.

Page 8, line 25: what is meant by "molecules with finite vapor pressures"? Isn't the vapor pressure finite with a given structure at a given temperature and pressure?

Page 10, line 7-9: is Sp the surface area of the particles? Sentence starting with "Varying Lp(HOM)" is confusing, isn't that relationship linear by definition? Consider rewording to replace the verb "led", similar to the sentence on page 13, lines 11-12.

Page 11 line 16-17: the comment about the endo versus exocyclic bonds should either be explained and/or cited here, or be moved to the discussion section. Fig 2: Should note what these were normalized to as in Figure 4 caption.

Page 13 line 11-13: is this based on assumptions or was new particle formation observed? Consider rewording for clarity. Does loss to particles include loss to NPF in the correction?

Fig 6: why does the C10H16O7-10 vary so much from the C10H14O7-10 & C10H16OxNO2 homologues in that oxygen range? Is it possible that the peaks are mis-assigned or contain multiple compounds? On page 13 line 22 it is stated that: "the potential to condense on particles was about the same for HOM-PP and HOM-ON", however this is not entirely consistent with C10H16O7-10 in Figure 6?

Page 16 paragraph starting line 15: Where does the factor of 1/5 come from? Also, I find the distinction, or lack thereof, of OrgNO3 and ON confusing throughout the manuscript but particularly in this paragraph. Please clearly define the difference somewhere or use the same acronym throughout.

Page 17 line 4-5 & 7-8: would be helpful to define HOM with respect to # of Oxygens or to define "few O-atoms" with a number. This could be defined on page 3, lines 13-14 or lines 33-34.

Page 18, line 9-10: do you see this higher SOA mass for NOx experiments relative to no-NOx? Has it been reported elsewhere?

Page 18, 1st paragraph section 4.3: can you describe better what Zhao did and do a more complete comparison? As-is, it seems like you're in perfect agreement that 10-11% of the OrgNO3 is in the particle phase, but then lines 18-19 says you're not in agreement?

Section 4.4: language could be formalized more throughout this section. "Mass loss" is a somewhat misleading phrase to mean converting potentially condensing species to non-condensable, not a direct loss of SOA mass due to, for example, evaporation. Please rework to be clearer and more streamlined.

page 21 line 12-13: please provide citations

---

## Referee Comment (RC3) · Anonymous Referee #3 · 20 Mar 2020

This manuscript reported HOM organic nitrates, permutation products, and accretion products formation from the oxidation (mostly OH oxidation) of a-pinene and b-pinene. Effective uptake coefficients of HOM on particles was also investigated and reported. Experiments were conducted in CSTR under high RH conditions without seed particles (except for the uptake experiments). It was found that increasing NOx affects the fraction of each type of HOM products formed. The fraction of organic bound nitrate (OrgNO3) stored in gas-phase HOM-ON was found to be substantially higher than the fraction of particulate OrgNO3 and was attributed to particle-phase hydrolysis

of OrgNO3. Lastly, SOA yields were also reported and discussed. The suppression of SOA yields with increasing NOx was attributed to suppression of gas-phase HOM accretion products.

This is an interesting study and falls within the scope of ACP. It contributes to our further understanding of monoterpene oxidation in the presence of NOx and the resulting organic nitrate formation and chemistry. There are three main comments that should be addressed prior to publication 1) more analysis needs to be conducted to reconcile the discrepancy between the fraction of particulate OrgNO3 reported in this study and other prior studies in literature. The authors attributed this to hydrolysis but this is not supported by data in literature, 2) more details need to be provided regarding the evaluation of the effective uptake coefficients, and 3) the manuscript should be edited for language. More detailed comments are provided below.

Specific comments

1. Page 5 line 12. It would be useful to include the amount of ozone added and the steady state ozone concentration in Table 1.

2. Page 6 line 16. Were the seed particles dried before being injected into the chamber? I would assume not but it is not clear from the manuscript. Please specify.

3. Page 8 section 2.4 a. Line 25. What does "finite vapor pressure" mean?

b. I do not fully understand how these experiments were conducted. From this section, it appears that experiments without seeds (and no organic aerosol formation via nucleation) were compared with experiments with seeds (ammonium sulfate particles injected) to determine the effective uptake coefficient. My understanding from the experimental section is that seed particles were only added in "experiment series 3" (i.e., experiments to determine effective uptake coefficient), and no seed particles were added in all other experiments. Presumably, organic aerosol formation via nucleation took place in all other experiments. However, according to Table 1, the b-pinene mixing ratio used in "experiment series 3" was the same as other all experiments. If so, shouldn't nucleation also took place in ""experiment series 3", and that particles (organic particles) would be present in the system even though no ammonium sulfate particles were added (see page 13 line 12)? If this is the case, one needs to consider uptake onto pure organic particles? Please describe and discuss these clearly in the revised manuscript.

4. Page 12, line 12. The authors noted that the highly-oxidized C<10 nitrates were observed with increasing NOx and that "supposedly, they did not arise from gas-phase chemistry but were formed at the walls". Please elaborate. What mechanisms at the walls? If there is chemistry on the walls, how would this affect section 2.4 (determination of uptake coefficient) if there is also some sort of wall memory?

5. Page 13, line 16, please also indicate (e.g., in Figure 7) the organic mass concentration, as SOA yield is also highly dependent on organic mass.

6. Page 14, lines 1-2. Are these mass concentrations in the gas and particle phases consistent with equilibrium partitioning of HNO3?

7. Page 16, line 9. There are many more studies. For example, see review and references in Ng et al. (ACP, 2017). Some more recent studies, for example, Claflin and Ziemann (J. Phys. Chem. A, 2018), are also relevant.

8. Page 16 line 16. A recent study by Takeuchi and Ng (ACP, 2019) also reported on the ON formed by photooxidation of monoterpenes.

9. Page 16 line 23. Please provide citations for this statement "We found contributions between 0 % and 2.7 % by AMS, which is within the range of most other data reported in the literature but at the lower end.".

10. Page 17 line 9. It was stated that "Our findings are in agreement with observations by Lee et al. (2016b) in a field study." My understanding is that the study by Lee et al. was conducted in a rural environment, presumably with very low level of NOx.

The ambient conditions were quite different from laboratory conditions employed in this study. Please justify why an agreement would be expected between results in Lee et al. and this study.

11. Page 18-19, section 4.3. The fraction of OrgNO3 is much lower in this study than Zhao et al. and other studies in literature. The authors attributed this to potential hydrolysis of organic nitrates in the particle phase as experiments in Zhao et al. were conducted at much lower RH. However, a recent study by Takeuchi and Ng (ACP, 2019), conducted at similar RH to this study, showed that the fraction of organic nitrates in the particles is also much higher than that reported in this study, and the fraction or organic nitrates undergoing hydrolysis was constrained. More analysis should be conducted here to evaluate why the value reported in this study is much lower than prior literature.

―――――――――――――――――――

---

## Author Comment (AC4) · 15 May 2020

We thank referee#1 for the helpful comments. Please, find our responses in the pdf-file attached. Please, see new Figures 3 & 4 below.

Please also note the supplement to this comment:
https://www.atmos-chem-phys-discuss.net/acp-2019-1168/acp-2019-1168-AC4-supplement.pdf

[Figure]

[Figure]

[Figure]

**Figure 3.** Mass concentration of HOM products in dependence on [NOₓ]ₛₛ in α-pinene photo-oxidation experiments. C₅-C₂₀ compounds with molecular masses 230-550 Da were added up for total HOM (black squares) and divided into HOM monomers (light blue circles) and HOM accretion products (blue triangles). The analysis is based on the assigned peaks (>90% of the total signal) and the sensitivity of $3.7 \times 10^{10}$ molecules cm⁻³ nc⁻¹ (suppl. section 1.2). HOM accretion products decrease with increasing [NOₓ]ₛₛ: at the lowest and highest NOₓ levels of 0.3 ppb and 72 ppb HOM-ACC contribute 0.3 µg m⁻³ and 0.09 µg m⁻³, respectively, to total HOM, whereas HOM monomers contribute about 0.4 µg m⁻³ over the whole range. More than 70% of HOM-ACC were suppressed at the highest [NOₓ] while HOM monomers remained about constant. The increasing importance of alkoxy radicals with increasing [NOₓ]ₛₛ is indicated by the small circles: C₅.₉ compounds (small open circles) arise in large parts from fragmentation of alkoxy radicals. They double from ≈0.9 to ≈1.8 µg m⁻³ at the highest [NOₓ]ₛₛ, whereas the C₁₀ compounds (grey circles) drop by only about 30%. C₅.₉ compounds must carry at least 7 O-atoms because the lower end of the mass range is set to 230 Da which is the molecular mass of C₁₀H₁₄O₆. Assuming that compounds in the selected mass range will contribute to SOA formation, the lower SOA yields at high [NOₓ] was due to the suppression of accretion products and increasing fragmentation via the alkoxy path played a minor role. Dashed and dotted lines save to guide the eye and have no further meaning. Concentrations were corrected as described in supplement section S1.2. Turnover ranged from $8.7 \times 10^7$ cm⁻³s⁻¹ and $1.04 \times 10^8$ cm⁻³s⁻¹ leading to correction factors in a range of 1.1 - 0.8. The correction factors were close to one thus did not add much uncertainty. Observed particle surface ranged from ~10⁻⁶ m²m⁻³ to $6 \times 10^{-5}$ m² m⁻³ resulting in correction factors between 1.0 and 1.45 with the highest correction factors at lower [NOₓ]ₛₛ where new particle formation could not be suppressed.

**Fig. 1.**

[Figure]

Figure 4: HOM pattern from α-pinene photo-oxidation at two $NO_X$ levels in the monomer range. Panel A: low $NO_X$ conditions ([α-pinene]$_{SS}$ = 1.7 ppb, [NOX]$_{SS}$ = 0.3 ppb), Panel B: high $NO_X$ conditions ([α-pinene]$_{SS}$ = 1.0 ppb, [NOX]$_{SS}$ = 8.7 ppb). Black bars: HOM-PP termination products of reactions R3 and R4a. Blue bars: HOM-ON (organic nitrates). Red bars = HOM-RO$_2$ (peroxy radicals). The signals were normalized to the sum over all detected ions. Panel C: Mass concentrations of HOM monomers (green) in the molecular mass range 230-550 Da. HOM-ON (blue) are increasing with increasing [NOX]$_{SS}$, HOM-PP (black) are decreasing, while the sum of all HOM-monomers remains about the same. At about 10 ppb [NOX]$_{SS}$ HOM-ON make up half of the HOM monomers and at 26 ppb [NOX]$_{SS}$ they make up about 50% of the total HOM (shown in Figure 3).

39

**Fig. 2.**

**Supplement:**

Reply to reviewer #1

We thank the reviewer #1 for the helpful comments. We addressed all points raised.

Reviewer comment:

1. The paper concluded with increasing NOX HOM-ACC strongly decreases and consequently suppress SOA formation. While the experiments and analysis appear robust and in agreement with some literature, it is important to point out that some other literature such as Pye et al., (2015) and Marais et al. (2016), with specific representation of particulate organic nitrate predict the reduction in NOx emissions causes a considerable reduction in organic aerosol. Could authors comment on this discrepancy? In this regard, in Figure 2 authors showed HOM spectra with and without NOX addition. It might be worthwhile to mention total SOA or HOM mass for these two cases for easier comparison.

Reply:
In our paper Safrazadeh et al. (2016) we showed that the effect of $NO_X$ on SOA yields is complex. $NO_X$ can affect SOA formation by influencing the OH concentration (a variation of OH scavenging as described in McFiggans et al. (2019)) and by changing the product composition. How the product composition and the product properties are changing with $NO_X$ has general components, such as NO reacting fast with $RO_2$, and is specific for the compound (class) under consideration.

Marais et al. (2016) propose an improved mechanism for isoprene oxidation and applied it in a regional model study. They showed that increasing $NO_X$ (NO) is leading to decreasing *isoprene* SOA: "Isoprene SOA concentrations increase as NOx emissions decrease (favoring the low-NOx pathway for isoprene oxidation)." Herein the low-$NO_X$ pathway leads to the formation of IEPOX, which in turn enter the particulate phase by reactive uptake. The reactive uptake of IEPOX is specific to isoprene reaction systems, and we don't really see what could be learned from that comparison for our study of monoterpenes.

The work of Pye et al. (2015) is also a regional model study considering the role of NO and $NO_3$ in organic nitrate (ON) formation. The results of regional model studies depend on parametrization of the precursor chemistry (which is in this case lumped), while we describe specific observations in our chamber. Pye et al. find a decrease of 9% in SOA when $NO_X$ emissions are reduced by 25%. The $NO_X$ level was roughly of the order of one 1 ppb, the effect of the $NO_X$ reduction on the $NO_X$ concentration was not specified. Pye et al. attributed the monoterpene-SOA decrease with decreasing $NO_X$ mainly to a decreasing $NO_3$ contribution. Effects of $NO_3$ are outside our observations, and we added a statement in the Introduction section, underling that our study focus on $NO_X$ in daytime photochemical systems and that $NO_3$ reactions also lead to ON, but are not treated here ==(p.2, line 16-20, in the revised manuscript).==

Since Marais et al. as well as Pye et al. address aspects different than our study, it is very difficult to say if their model results are in contradiction to ours or not. So, detailed specific comparisons with Marais et al. or Pye et al. do not make much sense to us. However, the referee is correct that our laboratory results, which address mechanistic aspects of SOA formation in the presence of $NO_X$ especially under consideration of HON-ON, should not be generalized too quickly and blindly transferred to the atmosphere, where more aerosol precursors and SOA formation processes prevail. In order to prevent misleading interpretations of our specific results we will add a sentence in the ==concluding section (p.23, line 26-31):==

"Note, that we considered the photochemistry of $NO_X$ to SOA contribution for two major MT, α-pinene and β-pinene. We find that that SOA yields are fairly independent of $NO_X$, but drop significantly at the highest $NO_X$ levels. Model studies show that increase of $NO_X$ emissions may also lead to more SOA, when $NO_3$ is the oxidant (e.g. Pye et al. 2015) or when isoprene is involved (Marais et al. 2016). In the latter case NO directs the gas phase mechanism toward isoprene products with reactive uptake, while for compounds like α-pinene and β-pinene, which were investigated here, condensation is more important for SOA formation and thus vapor pressures controls SOA yields."

We did not modify Figure 2 as this illustrates how the effect of $NO_X$ appears quite obvious in the mass spectra. Instead we modified Figure 3, which now shows mass concentrations of total HOM and of monomers and accretion products. These were derived from high resolution analysis of the mass spectra, as this allowed for a more detailed analysis. We replaced integration of mass spectra at UMR in certain ranges by analysis based on peak lists (Suppl. S6). We can thus resolve monomers and accretion products in the overlapping range. Identified peaks explain more than 90% of the observed signal. We also replaced mixing ratio by mass concentration as this allows for a better direct estimate how much mass monomers and accretion products potentially contribute to SOA. We limited the analysis to the mass range 230-550Da, which covers the compounds with sufficient functionalization to condense and form SOA. Therefore the numbers are somewhat lower compared to the original manuscript, but this did not affect any of the conclusions.
With new Figure 3 we also modified formulations in section 3.3 (p.12, line 26 – p.13, line12, in the revised manuscript).

2. Page 11, Line 2
a) The authors estimated a molar yield of ≈36% for the ON formed from β–pinene which is higher than the largest previously reported values (26 ±7% Rindelaub et al., (2016), while for similar condition of this study (acidic seed aerosol and RH ≈60%), they even estimated less (≈6%).
b) The authors mentioned at the Figure 9's caption "The effect of hydrolysis of 80% of the organic bound nitrate has no substantial effect on the SOA mass." However, Boyd et al. (2015) estimated particle phase hydrolysis of organic nitrates compose 45–74% of the organic aerosol. These discrepancies might be attributed to estimation of a slower aerosol hydrolysis in this study? and subsequently underestimation of importance of hydrolysis for explaining the SOA mass suppression with increasing NOX in the system?

Reply:
a)      Our molar yield considers the sum of all organic nitrates formed from β-pinene in the gas-phase in absence of seed aerosol at RH = 60%. Our finding is commensurable with molar branching ratios of the reaction $RO_2$ + NO into RO and ON as implemented in MCMv3.1.1 for ß-pinene for the given conditions. Since we did not have seed aerosols in this experiment, the observed molar yield cannot be dependent on the pH of the seed aerosols. ON formation depends of course both on the VOC under investigation and on the VOC/NOX ratio. Therefore one cannot expect the same yields in different studies with different precursor starting conditions.

Rindlaub et al. (2015) investigated α-pinene and reported an apparent yield for ON of 26(±7)% by extrapolation to particle free conditions. Assuming that α-pinene and β-pinene have a similar ON/RO branching ratio in the reaction of $RO_2$ with $NO/NO_2$ this value is within the errors the same as our 36(±4)%. While Rindelaub et al. observed the integral ON by actively measuring the available ON by FTIR, we derive our value from the consumption of $NO_2$, more precisely from the excess of $NO_2$ in the absence of β-pinene at the same OH concentration. So, as long as the branching ON/RO does not much depend on RH, which is the case, we would observe by our method the same apparent ON yield at all RH, while Rindelaub et al. would miss the hydrolysed ON. It is not on us to judge Rindelaub's et al. work, but it was performed for another compound, α-pinene, at ppm initial concentrations and more than 100ppb of $NO_X$ initially in the system. Moreover, there were several hundred to several thousand ug/m$^3$ particulate mass formed during their non-seeded experiments, so a substantial fraction of the ON, including semivolatile ON, must have resided in the particulate phase. In our interpretation of their Figure 4 they observed overall a yield of ON of 5-10% for a wide range of RH with strong excursions up to 23% in a narrow range around 15% RH. We reformulated a part of section 3.5 to include references proposed by the reviewer (p.17, line 8-13).

b)      The intent of our study is clearly *not* to solve the question of hydrolysis of particulate ON. Our starting point of discussing HOM-ON hydrolysis is to find a rationale for the mismatch between observed $OrgNO_3$ (particulate organic bound nitrate) observed by AMS (up to 3%) and the $OrgNO_3$ which should be expected from the uptake/condensation of HOM, including HOM-ON, with more than 6 O (up to 10%). The mismatch indicated that more than 2/3 of the $OrgNO_3$ got somehow lost. Nevertheless, as shown in Figure 9, the role of hydrolysis for the impact of $NO_X$ for SOA production is negligible. Since this has been proposed in the literature, we were merely wondering if hydrolysis of ON in the particulate phase may help to reconcile particulate phase observations and HOM-ON.

Our Figure 6 shows that vapor pressures of HOM do not depend much on the type of the termination group. It seems realistic to us that only $HNO_3$ will escape the particulate phase on hydrolysis of multi-functionalized HOM-ON, while the multi-functionalized organic moieties remain. Under this specific case/assumption we calculated the change of the expected mass as shown in Figure 9; it assumes in fact instantaneous, thus fast hydrolysis. The release of $HNO_3$ can explain the mismatch between gas phase $OrgNO_3$ that is expected to condense on particles and realized $OrgNO_3$ in the particulate phase. Figure 9 is supposed to show that this type of hydrolysis will not much affect the SOA mass thus our finding of a low dependence of SOA yield on $NO_X$, if *OH concentration is kept stable*. The somewhat lower yields at high $NO_X$ in our case can be solely accounted for by less mass formation due to HOM by suppression of dimers with increasing $NO_X$. Overall, our interpretation is not in contradiction with field and model studies which detected that about 2/3 of the particulate ON hydrolyze and release $HNO_3$ from the particulate phase (Zare et al., 2019, Fisher et al., 2016). Like Takeuchi and Ng (2019), these studies leave the fate of the organic moieties open, but they are likely alcohols (Hu et al. 2011) and thus have the same or lower vapor pressures as the mother ON (Zare et al., 2019).

We are aware of and appreciate the study by Boyd et al. 2015. However here must be a misunderstanding. First of all, Boyd et al. investigated the oxidation of β-pinene by $NO_3$. Herein $NO_3$ is the primary oxidant forming nitrate groups directly by first attack on the double bond. This is different from our study where we oxidize the MT by OH and (HOM-)ON are formed by peroxy radicals terminating with NO and $NO_2$. Boyd et al. found 45-74% contribution of particulate ON of which 90% survived hydrolysis, due to a lower fraction of tertiary ON in $NO_3$ oxidation compared to photooxidation and termination by $NO_X$ (our case).

Browne et al. (2013) suggested that photooxidation of α-pinene and termination by $NO_X$ produces about 60% tertiary ON that easily hydrolyze, but they considered ON hydrolysis only for $HNO_3$ budget. The suggestion by Browne et al. would be perfectly in line with our interpretation, that about 2/3 of the $OrgNO_3$ is lost as $HNO_3$. Takeuchi and Ng (2019) stated in a recent paper that they cannot determine for sure the "fate of the organic moiety of the hydrolysis product (i.e., stay in the particle phase or repartition back to the gas phase)". The hydrolysis was observed by Takeuchi and Ng for about the same relative humidity. The hydrolyzable fraction FH for α-pinene +OH + NO system was 32%, if one assumes no loss of the organic moiety. A hydrolyzed fraction of 32% is about half what we determined, but regarding our only rough estimates it is within the errors. The hydrolysis was fast and we are able to easily observe it within the residence time of 1h in our chamber. Taken all together the findings of the Boyd et al, Browne et al. (2013), and Takeuchi and Ng l. are supportive or at least not in contradiction to our findings. Furthermore, the experiment of Zhao et al. (2018) showed that expected $OrgNO_3$ and observed $OrgNO_3$ agree in a dry environment. These experiments were made in another context but with the same instruments and analysis methods. Together with most of the literature data this all is in favor of our interpretations. However, we would like to note that hydrolysis of ON and pON is not the focus of the paper and our data are not suited to solve inconclusive findings in the literature.

3) Page 18, Line 11-24 The authors discussed higher humidity in their chamber and hydrolysis as the key for less estimation of $OrgNO_3$ fraction in the particulate-phase than as determined by AMS and also finding ($\approx$11%) by Zhao et al. (2018). It is important to point out not only Zhao et al. but also many recent measurements (e.g., Romer et al., 2016) and modeling studies (e.g., Pye et al., 2015; Fisher et al., 2016; Zare et al., 2019) estimated a higher fraction of organic nitrates in the particle phase ($\approx$10%-20%). As they also considered a rather fast hydrolysis for organic nitrate aerosols it might be worthwhile to compare the result here to their results as well. However, it might be useful to mention Zare et al. show that at a more humid condition (similar to this study with higher RH) heterogeneous uptake to particle water tends to form less particulate organic nitrates against uptake to dry organic aerosols. Considering the impact of humidity at the aerosol formation together with the impact on the loss process of hydrolysis for particulate organic nitrates could help reconcile the discrepancy?

Reply:
We think here is a misunderstanding. As was shown in Figure 8 (blue and orange circles) the molar fraction of particulate ON expected from HOM condensation ranges up to 40% for both MT with increasing $NO_X$. In the new Figure 4 we show now that the mass fraction of HOM-ON mounted up to 50%. From this point of view we are cum granis sale in agreement with the other lab, field and model studies which find similar fractions of particulate ON from MT. The diagnostic link between HOM-ON and particulate phase analysis by AMS was organic bound nitrate ($OrgNO_3$), a terminus used by the AMS community. We discuss $OrgNO_3$, i.e. the mass fraction of $-NO_3$ groups attached to ON. In Figure 8 we showed the $OrgNO_3$ mass fraction expected from condensable HOM-ON and compare it with the $OrgNO_3$ fraction observed by AMS. We tried to clarify the difference between $OrgNO_3$ and particulate ON throughout the manuscript, e.g. we added a note to section 2.3 (p.8, line 16-17, in the revised

manuscript), reformulated large parts of section 4.1. (p.18, line 2-15, in the revised manuscript) and (p.18, line 22-23, in the revised manuscript).

We attributed the discrepancy between $OrgNO_3$ in the gas-phase HOM and in the particle phase to plausible hydrolysis of ON in the particulate phase (as discussed in the previous reply). Since we are missing organic bound $NO_3$ in the system and *not* SOA mass, we assumed that only $HNO_3$ is released to the gas-phase. As shown in Figure 9 this effect is small, because $OrgNO_3$ contributed only about 20% to HOM-ON mass and HOM-ON contributes a few times 10% percent to SOA mass.

The HOM-ON hydrolysis in the particulate phase must occur on times scales of less than one hour at 60%RH. The time scale is in accordance with the findings of Romer et al. (2016), however they linked ON hydrolysis to isoprene oxidation, which we don't have. We choose Zhao et al. as an example for expected $OrgNO_3$ and observed $OrgNO_3$ being in agreement in a dry environment, because they used the same instruments and analysis methods.

We determined effective uptake rates for HOM-ON and the $OrgNO_3$ content in the particulate phase at only one relative humidity. We did not determine effective uptake rates for HOM ON at different humidity and therefore our data do not allow statements on the effects of humidity on particle formation by condensation of HOM.

Although hydrolysis of ON is very interesting and the most plausible explanation for the loss of $OrgNO_3$, it is not a central point of out studies and we prefer to refrain from further discussion in the manuscript.

We modified Figure 4 substantially in order to show the mass fraction of HOM-ON.

We added the references (see below) for discussion here to manuscript.

Minor comments
We are sorry for that many mistakes and typos. We thank the reviewer for the careful reading of the manuscript and for the corrections. We went through the manuscript in order to improve language and grammar.

Page 3, for less confusion and similar to the other relevant papers, it is better to give a same reaction number for reactions with similar reactants, e.g. (R4) and (R4a) should be "(R4a)" and "(R4b)", and also for R5 and R7 should change as "(R5a)" and "(R5b)".
done

Page 6, Line 1-4, multiplication sign for reaction rates are missed.
dots replaced by x

Page 6, Line 2, References should be lined up in the proper sequence.
done

Page 13, Line 13, remove spare space before parentheses.
done

Page 15, Line 1-4, remove redundant parentheses.
We could not find redundant parenthesis. The parenthesis being there were set by purpose to indicate a side remark.
No action.

Page 15, Line 21, remove extra "was"
done

Page 15, line 24, "estimated to be"
done

Page 32, Figure1, for better readability write axis label on the right-hand side from
done, for consistency also applied to Figure 7

Page 37, Line4, the brown bars look like more "orangeish" than brown in my eyes.
color tuned to brown

Page 40, Line 5-6 used different font.
corrected

References

Boyd, C. M., Sanchez, J., Xu, L., Eugene, A. J., Nah, T., Tuet, W. Y., Guzman, M. I., and Ng, N. L.: Secondary organic aerosol formation from the β-pinene+NO3 system: effect of humidity and peroxy radical fate, Atmos. Chem. Phys., 15, 7497-7522, 10.5194/acp-15-7497-2015, 2015.

Browne, E. C., Min, K. E., Wooldridge, P. J., Apel, E., Blake, D. R., Brune, W. H., Cantrell, C. A., Cubison, M. J., Diskin, G. S., Jimenez, J. L., Weinheimer, A. J., Wennberg, P. O., Wisthaler, A., and Cohen, R. C.: Observations of total RONO2 over the boreal forest: NOx sinks and $HNO_3$ sources, Atmos. Chem. Phys., 13, 4543-4562, 10.5194/acp-13-4543-2013, 2013.

Fisher, J. A., Jacob, D. J., Travis, K. R., Kim, P. S., Marais, E. A., Chan Miller, C., Yu, K., Zhu, L., Yantosca, R. M., Sulprizio, M. P., Mao, J., Wennberg, P. O., Crounse, J. D., Teng, A. P., Nguyen, T. B., St. Clair, J. M., Cohen, R. C., Romer, P., Nault, B. A., Wooldridge, P. J., Jimenez, J. L., Campuzano-Jost, P., Day, D. A., Hu, W., Shepson, P. B., Xiong, F., Blake, D. R., Goldstein, A. H., Misztal, P. K., Hanisco, T. F., Wolfe, G. M., Ryerson, T. B., Wisthaler, A., and Mikoviny, T.: Organic nitrate chemistry and its implications for nitrogen budgets in an isoprene- and monoterpene-rich atmosphere: constraints from aircraft (SEAC4RS) and ground-based (SOAS) observations in the Southeast US, Atmos. Chem. Phys., 16, 5969-5991, 10.5194/acp-16-5969-2016, 2016.

Hu, K. S., Darer, A. I., and Elrod, M. J.: Thermodynamics and kinetics of the hydrolysis of atmospherically relevant organonitrates and organosulfates, Atmos. Chem. and Phys., 11, 8307-8320, 10.5194/acp-11-8307-2011, 2011.

Marais, E. A., Jacob, D. J., Jimenez, J. L., Campuzano-Jost, P., Day, D. A., Hu, W., Krechmer, J., Zhu, L., Kim, P. S., Miller, C. C., Fisher, J. A., Travis, K., Yu, K., Hanisco, T. F., Wolfe, G. M., Arkinson, H. L., Pye, H. O. T., Froyd, K. D., Liao, J., and McNeill, V. F.: Aqueous-phase mechanism for secondary organic aerosol formation from isoprene: application to the southeast United States and co-benefit of SO2 emission controls, Atmos. Chem. Phys., 16, 1603-1618, 10.5194/acp-16-1603-2016, 2016.

Pye, H. O. T., Luecken, D. J., Xu, L., Boyd, C. M., Ng, N. L., Baker, K. R., Ayres, B. R., Bash, J. O., Baumann, K., Carter, W. P. L., Edgerton, E., Fry, J. L., Hutzell, W. T., Schwede,

    D. B., and Shepson, P. B.: Modeling the Current and Future Roles of Particulate
    Organic Nitrates in the Southeastern United States, Environmental Science &
    Technology, 49, 14195-14203, 10.1021/acs.est.5b03738, 2015.

Rindelaub, J. D., McAvey, K. M., and Shepson, P. B.: The photochemical production of
    organic nitrates from alpha-pinene and loss via acid-dependent particle phase
    hydrolysis, Atmos. Environ., 100, 193-201, 10.1016/j.atmosenv.2014.11.010, 2015.

Romer, P. S., Duffey, K. C., Wooldridge, P. J., Allen, H. M., Ayres, B. R., Brown, S. S.,
    Brune, W. H., Crounse, J. D., de Gouw, J., Draper, D. C., Feiner, P. A., Fry, J. L.,
    Goldstein, A. H., Koss, A., Misztal, P. K., Nguyen, T. B., Olson, K., Teng, A. P.,
    Wennberg, P. O., Wild, R. J., Zhang, L., and Cohen, R. C.: The lifetime of nitrogen
    oxides in an isoprene-dominated forest, Atmos. Chem. Phys., 16, 7623-7637,
    10.5194/acp-16-7623-2016, 2016.

Takeuchi, M., and Ng, N. L.: Chemical composition and hydrolysis of organic nitrate aerosol
    formed from hydroxyl and nitrate radical oxidation of α-pinene and β-pinene, Atmos.
    Chem. Phys., 19, 12749-12766, 10.5194/acp-19-12749-2019, 2019.

Zare, A., Fahey, K. M., Sarwar, G., Cohen, R. C., and Pye, H. O. T.: Vapor-Pressure
    Pathways Initiate but Hydrolysis Products Dominate the Aerosol Estimated from
    Organic Nitrates, ACSEarthSpaceChem., 3, 1426-1437,
    10.1021/acsearthspacechem.9b00067, 2019.

---

## Author Comment (AC5) · 15 May 2020

We thank referee#2 for the helpful comments. Please, find our responses in the pdf-file attached. Please, see new Table 1 and new Figures 3 below.

Please also note the supplement to this comment:
https://www.atmos-chem-phys-discuss.net/acp-2019-1168/acp-2019-1168-AC5-supplement.pdf

[Figure]

**Tables**

**Table 1: Overview of α-pinene and β-pinene experiments**

5

| Experiment Description | $[VOC]_0^a$ [ppb] | $[NO_x]_0^a$ & $([NO_x]_{ss}^b)$ [ppb] | $[O_3]_{ss}^b$ [ppb] | $[OH]_{ss}^b$ $[10^7 cm^{-3}]$ |
|---|---|---|---|---|
| 1. Gas-phase yield of ON and gas-phase OrgNO$_3$ (Section 3.1) | β-pinene  39→0
m-xylene  3.7 | 50
(20→30) | 19→30 | 2.3±20% |
| 2. Formation of HOM-ON (Section 3.3) | α-pinene  16.5 | 0.3     / 7.5     / 15.3$^c$ / 26.7 / 39.7 / 45.5
(0.3     / 1.8     / 3.7$^c$ / 5.7    / 8.7   / 10.4)
/ 52.9   / 59.1   / 83.3 / 137.8
(/ 12.4 / 15.8   / 26.8 / 72.2) | 62 -152 | 4.5 -7.5 |
|  | β-pinene  37 | 3.9   / 53.8  / 113.6 / 194
(1.2   / 16.5  / 37.0   / 77.) | Not determined | Not determined |
| 3. Effective uptake coefficients$^d$ (Section 3.4) | α-pinene  12.5 | 0.3
(0.3) | 29 | 9.2±20% |
|  | β-pinene  37 | 30
(4) | 49 | 8.8±20% |
| 4. OrgNO$_3$ in SOA (Section 3.5) | α-pinene  46 | 0.3   / 32.0  / 51.0  / 60.0
(0.3 / 10.4  / 17.5  / 19.5) | 37 - 62 | 4.7- 7.7 |
|  | β-pinene  38 | 0.3   / 6.7  / 13.4  / 32.9  / 54.8  / 103
(0.3 / 5.1  / 9.5    / 21.7  / 35.5  / 45.7) | 44 – 53 | 0.9 - 3.7 |

a  subscript $_0$ refers to mixing ratio in the inflow
b  subscript $_{ss}$ refers to mixing ratio in steady state
c  average of two experiments at $[NO_x]_0$ of 15 and 15.5 ppb ($[NO_x]_{ss}$ of 3.6 and 3.75 ppb)
d  in presence of ammonium sulfate seed aerosols

**Fig. 1.**

[Figure]

**Figure 3.** Mass concentration of HOM products in dependence on $[NO_X]_{SS}$ in α-pinene photo-oxidation experiments. $C_5$-$C_{20}$ compounds with molecular masses 230-550 Da were added up for total HOM (black squares) and divided into HOM monomers (light blue circles) and HOM accretion products (blue triangles). The analysis is based on the assigned peaks (>90% of the total signal) and the sensitivity of $3.7 \times 10^{10}$ molecules cm$^{-3}$ nc$^{-1}$ (suppl. section 1.2). HOM accretion products decrease with increasing $[NO_X]_{SS}$: at the lowest and highest $NO_X$ levels of 0.3 ppb and 72 ppb HOM-ACC contribute 0.3 μg m$^{-3}$ and 0.09 μg m$^{-3}$, respectively, to total HOM, whereas HOM monomers contribute about 0.4 μg m$^{-3}$ over the whole range. More than 70% of HOM-ACC were suppressed at the highest $[NO_X]$ while HOM monomers remained about constant. The increasing importance of alkoxy radicals with increasing $[NO_X]_{SS}$ is indicated by the small circles: $C_{5-9}$ compounds (small open circles) arise in large parts from fragmentation of alkoxy radicals. They double from ≈0.9 to ≈1.8 μg m$^{-3}$ at the highest $[NO_X]_{SS}$, whereas the $C_{10}$ compounds (grey circles) drop by only about 30%. $C_{5-9}$ compounds must carry at least 7 O-atoms because the lower end of the mass range is set to 230 Da which is the molecular mass of $C_{10}H_{14}O_6$. Assuming that compounds in the selected mass range will contribute to SOA formation, the lower SOA yields at high $[NO_X]$ was due to the suppression of accretion products and increasing fragmentation via the alkoxy path played a minor role. Dashed and dotted lines save to guide the eye and have no further meaning. Concentrations were corrected as described in supplement section S1.2. Turnover ranged from $8.7 \times 10^7$ cm$^{-3}$s$^{-1}$ and $1.04 \times 10^8$ cm$^{-3}$s$^{-1}$ leading to correction factors in a range of 1.1 - 0.8. The correction factors were close to one thus did not add much uncertainty. Observed particle surface ranged from ~$10^{-6}$ m$^2$m$^{-3}$ to $6 \times 10^{-5}$ m$^2$ m$^{-3}$ resulting in correction factors between 1.0 and 1.45 with the highest correction factors at lower $[NO_X]_{SS}$ where new particle formation could not be suppressed.

**Fig. 2.**

**Supplement:**

Reply to reviewer #2

We thank the reviewer #2 for the helpful comments. We addressed all points raised.

**Comments of Referee #2**

The goal of this work is to explore the observed and documented suppression of SOA formation in the presence of NOx from previous studies. The authors hypothesized the observed decrease in SOA mass could be due to HOM termination reactions, in particular the reactions that form gas-phase accretion products or HOM organic nitrates. To test their hypothesis, they performed a series of chamber experiments at the Jülich Plant Atmosphere Chamber, a well characterized environment. They primarily used a Chemical Ionization Mass Spectrometer with NO3- ionization which has been repeatedly shown to be useful in measuring HOM, and studied the photooxidation of _and _-pinene. They found that HOM accretion products were much more heavily suppressed than non-nitrate monomer products in the presence of NOx. They determine this loss of accretion products is a possible and even likely reason for SOA mass suppression in the presence of NOx.

Section 4.4 discusses the suppression of accretion products but notably lacking is the possibility of precursor decomposition as a result of RO2 + NO (R7, page 3 line 24) forming an alkoxy which can rapidly decompose, resulting in smaller RO2 building blocks for accretion chemistry. This is brought up briefly on page 11 line 23, but is not worked into the discussion in section 4.4. It is however noted in the last sentence of the manuscript, only to say that it will be discussed in a further manuscript. This seems warranted to be discussed within this manuscript, and as-is I find it to be a major deficit of this work that should be discussed somewhere, potentially section 4.4.

Reply:

The referee is of course correct, reaction $RO_2$ + NO leads to alkoxy radicals RO. However, fragmentation is not the only reaction path of RO.

Alkoxy radicals can react with $O_2$ and under H-abstraction and formation of carbonyl compounds. If the alkoxy radical is a HOM-RO then the product will still contribute to SOA. However, reaction with $O_2$ is the major path for small peroxy radicals, although Finlayson Pitt and Pitts in their textbook listed substantial branching ratios for this path, p. 190)

RO radicals fragment often by α-scission. In ring systems, like the pinenes, this can lead to ring opening keeping the back bone intact. If the alkoxy radical that breaks apart is a HOM alkoxy and the larger fragment has 7, 8, or 9 C atoms, they could still contribute to SOA formation.

In addition, RO can rearrange very fast by H-abstraction forming an OH group and a radical, which can continue the radical chain. If the H is abstracted from a C atom, $O_2$ can add to the carbon centered radical and the resulting peroxy radical can continue autoxidation.

If H is abstracted from a hydroperoxide group, then the new peroxy radical will be formed in a unimolecular step. Both rearrangement pathways will contribute to SOA formation.

And autoxidation - maybe in combination with the alkoxy pathway - is fast, as can be seen by the high degree of oxidation of the detected HOM-ON.

The referee is correct that fragmentation of alkoxy radicals could lower the SOA yield, however, although fragmentation occurs this does not affect the condensable mass. This can now be seen in the modified Figure 3. We separated the total HOM into monomers and

accretion products and specified for the monomers the contribution of $C_{10}$ compounds and $C_{<10}$ compounds as f[NOX]$_{SS}$. Figure 3 is limited to a mass range where compounds will condense according to our analysis (section 4.2, Figure 6). We assume that $C_{<10}$ compounds arise mainly from alkoxy radical decomposition. The importance of $C_{<10}$ compounds increases with $NO_X$, but they must carry more oxygen to get molecular masses larger than 230 Da, i.e.one O atom more per C atom lost. Therefore the $C_{<10}$ compounds will still contribute to the SOA mass. As a consequence, the major reduction of HOM mass at high [NOX]$_{SS}$ is due to the suppression of HOM-ACC, as can be seen in new Figure 3.
Our reference to a next paper dealing with alkoxy radicals in the conclusion section referred to the mechanistic aspects of HOM formation.

Action: We provided an extended Figure 3, and modified and extended the text in section 3.3 accordingly (p.12, line 26 –p.13, line12 in the revised manuscript). We further added text passages to section 4.4 (p.20, line 32 – p.21, line 6, in the revised manuscript) where we discuss the role of fragmentation via alkoxy radicals on HOM and SOA yield. We added one sentence to the conclusion section (p.23, line 23-25, in the revised manuscript).

Additionally, the grammatical errors throughout impede understanding of the manuscript and need significant improvement before publication.

Reply:
We apologize for that many mistakes and typos. We thank the reviewer for the careful reading of the manuscript and for the corrections. We went through the manuscript in order to improve language and grammar.

All in all, this work is novel and of interest to the readers of ACP, so after addressing these major concerns and the other minor technical edits detailed below this manuscript should be suitable for publication.

Page 1, line 33-35: This could also be due to the higher vapor pressure of HOMON relative to similar non-nitrates as you reference on page 17 line 26-27. Consider rewording to make clear you're basing this statement off the results in this paper.

Thanks for that hint. We clarified, that we are referring to our results of effective uptake of HOM-ON (p.1, line 33-35, in the revised manuscript).

Page 2, line 28: "the absence of particles in the presence of NOx": consider rewording. The presence of NOx doesn't mean there's no particles around.

Sentence modified (p.3, line 1-2, in the revised manuscript).

Page 5 line 28 – Page 6 line 7: It would be helpful to include the average or typical concentration of OH more clearly. It is state on Page 6 line 3 but it feels a bit buried and is a single value as opposed to the range stated for O3 and shown for OH in Fig.1. Furthermore, the range of O3 stated on page 5 line 13 does not match that on page 6 line 2. I would suggest adding the O3 and OH concentrations to Table 1.

Table 1 extended as requested.

Page 8, line 10: please justify ignoring wall loss by providing a reference for particle- wall loss rates on similar chambers, or a loss rate estimate from your measurements.

We added a reference to an earlier study: Mentel et al. 2009, lifetime of particles ≈ residence time of air in the chamber.

Page 8, line 25: what is meant by "molecules with finite vapor pressures"? Isn't the vapor pressure finite with a given structure at a given temperature and pressure?

reformulated at both instances
Page 10, line 7-9: is Sp the surface area of the particles? Sentence starting with "Varying Lp(HOM)" is confusing, isn't that relationship linear by definition? Consider rewording to replace the verb "led", similar to the sentence on page 13, lines 11-12.

now: surface area added
We reformulated the passage according the suggestions (p.10, line 25-29, in the revised manuscript)

Page 11 line 16-17: the comment about the endo versus exocyclic bonds should either be explained and/or cited here, or be moved to the discussion section. Fig 2: Should note what these were normalized to as in Figure 4 caption.

Since we are referring to well-known properties of compounds with exocyclic and endocyclic double bonds, we will not move that to discussions. We reformulated the text passage. The caption of Figure 2 was changed accordingly (p.12, line 7-17, in the revised manuscript)

Page 13 line 11-13: is this based on assumptions or was new particle formation observed? Consider rewording for clarity. Does loss to particles include loss to NPF in the correction?

The formation of new particle does not matter much for the uptake experiments themselves as long as the particle surface area is determined correctly. However, one needs particle free conditions to measure directly the reference $c^0(HOM)$ at the same HOM production rate. The wall loss coefficient $L_W$ can be determined at lower concentrations. Too low $c^0(HOM)$ because of NPF will cause negative intercept in $L_P$ vs. $S_P$ plots like Figure 5. We modified the manuscript and explained more precisely what was done to determine $c^0(HOM)$ despite of NPF during the non-seeded part of the experiment (p.10, line 31 – p.11, line 2, and p.14, line17-22, in the revised manuscript).

Fig 6: why does the C10H16O7-10 vary so much from the C10H14O7-10 & C10H16OxNO2 homologues in that oxygen range? Is it possible that the peaks are mis-assigned or contain multiple compounds? On page 13 line 22 it is stated that: "the potential to condense on particles was about the same for HOM-PP and HOM-ON", however this is not entirely consistent with C10H16O7-10 in Figure 6?

We checked and the peaks are not miss-assigned. And yes, more than one compound can contribute to a given molecular formula. For example, a compound with molecular structure that causes very low vapor pressure could dominate the signal. We added a short discussion of that issue to the manuscript (p.19, line 4-5, in the revised manuscript).

Page 16 paragraph starting line 15: Where does the factor of 1/5 come from? Also, I find the distinction, or lack thereof, of OrgNO3 and ON confusing throughout the manuscript but particularly in this paragraph. Please clearly define the difference somewhere or use the same acronym throughout.

The factor 1/5 is the mass fraction of nitrate ($NO_3^-$, 62 Da) in ON with an average molecular mass of 300 Da. We clarified that.
The referee is correct, there are places where $OrgNO_3$ and ON were mixed. Otherwise $OrgNO_3$ is a terminus technicus used by the AMS community and we would like to stick to it. We defined $OrgNO_3$ clearly in the AMS section, and underlined its importance as diagnostic quantity for linking particulate phase observations and gas-phase observations.
We went through manuscript and tried to clearer distinguish ON and $OrgNO_3$. We added a note to section 2.3 (p.8, line 16-17, in the revised manuscript), reformulated section 4.1. in large parts (p.18 line 2-15, in the revised manuscript) and modified the first paragraph of section 4.2 (p.18, line 22-23, in the revised manuscript).

Page 17 line 4-5 & 7-8: would be helpful to define HOM with respect to # of Oxygens or to define "few O-atoms" with a number. This could be defined on page 3, lines 13-14 or lines 33-34.

We specified the number of Oxygen numbers as requested
We added the definition of HOM in the introduction section (p.4, line 12-13, in the revised manuscript).

Page 18, line 9-10: do you see this higher SOA mass for NOx experiments relative to no-NOx? Has it been reported elsewhere?

This effect will be very difficult to determine. At SOA yields of 10-20% with 50% pON it would be a 1 % effect. It might be overlapped by effects of varying [OH] and the HOM-ACC suppression. It has thus not been reported elsewhere. More importantly, hydrolysis of ON in the particulate phase will probably mask the effect. That is what we tried to address.
We reformulated this part of the manuscript to indicate that the effect would be small (p.19, line 29-31, in the revised manuscript).

Page 18, 1st paragraph section 4.3: can you describe better what Zhao did and do a more complete comparison? As-is, it seems like you're in perfect agreement that 10-11% of the OrgNO3 is in the particle phase, but then lines 18-19 says you're not in agreement?

We added more information of the Zhao experiments (p.20, line 3-7, in the revised manuscript).
Zhao et al. find agreement between expected $OrgNO_3$ from HOM and observation in the particulate phase, whereas in our study we only realized 20-30% of the expected $OrgNO_3$

from HOM in the particulate phase. We expressed clear wherein the studies do agree and do not agree (p.20, line 8-10, in the revised manuscript).

Section 4.4: language could be formalized more throughout this section. "Mass loss" is a somewhat misleading phrase to mean converting potentially condensing species to non-condensable, not a direct loss of SOA mass due to, for example, evaporation. Please rework to be clearer and more streamlined.

Paragraph was reformulated in large parts. The notation mass loss was omitted, where possible. (p.21, line 5 – p. 22, line 8, in the revised manuscript)

page 21 line 12-13: please provide citations

references provided

References

Mentel, T. F., Wildt, J., Kiendler-Scharr, A., Kleist, E., Tillmann, R., Dal Maso, M., Fisseha, R., Hohaus, T., Spahn, H., Uerlings, R., Wegener, R., Griffiths, P. T., Dinar, E., Rudich, Y., and Wahner, A.: Photochemical production of aerosols from real plant emissions, Atmospheric Chemistry and Physics, 9, 4387-4406, 2009

---

## Author Comment (AC6) · 15 May 2020

Reply to reviewer #3

We thank the reviewer #3 for the helpful comments. We addressed all points raised.

**Anonymous Referee #3**

This manuscript reported HOM organic nitrates, permutation products, and accretion products formation from the oxidation (mostly OH oxidation) of a-pinene and b-pinene. Effective uptake coefficients of HOM on particles was also investigated and reported. Experiments were conducted in CSTR under high RH conditions without seed particles (except for the uptake experiments). It was found that increasing NOx affects the fraction of each type of HOM products formed. The fraction of organic bound nitrate (OrgNO3) stored in gas-phase HOM-ON was found to be substantially higher than the fraction of particulate OrgNO3 and was attributed to particle-phase hydrolysis of OrgNO3. Lastly, SOA yields were also reported and discussed. The suppression of SOA yields with increasing NOx was attributed to suppression of gas-phase HOM accretion products.

This is an interesting study and falls within the scope of ACP. It contributes to our further understanding of monoterpene oxidation in the presence of NOx and the resulting organic nitrate formation and chemistry. There are three main comments that should be addressed prior to publication
1) more analysis needs to be conducted to reconcile the discrepancy between the fraction of particulate OrgNO3 reported in this study and other prior studies in literature. The authors attributed this to hydrolysis but this is not supported by data in literature,
2) more details need to be provided regarding the evaluation of the effective uptake coefficients, and
3) the manuscript should be edited for language. More detailed comments are provided below.

Reply:
1)      In contrast to the opinion of referee #3 we are convinced that literature data is in support of our explanations. We would like to note that hydrolysis of ON and particulate ON is not the focus of our paper and our data at one RH are not suited to reconcile possibly inconclusive findings in the literature:
a)      Our methodology is sound: because the effective uptake coefficients for both, HOM-PP and HOM-ON are close to unity, we can predict the mass fraction of ON and also organic bound nitrate ($OrgNO_3$) in particles. $OrgNO_3$ expected by uptake of HOM was compared to the amount of $OrgNO_3$ found in the particles by direct measurement. We found a strong discrepancy. A likely explanation for this difference is hydrolysis of organic nitrates in the condensed phase, as this is discussed extensively in the literature (e.g.). The experiments here were performed at 60% RH. Considering the strong dependence of hydrolysis on relative humidity we could also explain the differences of the $OrgNO_3$ content in particles found by us in different experiments at lower relative humidity of 30% (here and Zhao et al., 2018).
b)      The production of ON depends on the VOC/NOX ratio and only a fraction of the ON will be transferred to the particulate phase and contribute to SOA. Hydrolyzing of the

particulate phase ON depends on their structure (Boyd et al. 2015, Browne et al. 2013, Hu et al.2011), relative humidity, and the acidity of the particles (Rindelaub et. al. 2015). Hence, the fraction of particulate $OrgNO_3$ in particles can be variable in lab studies at different conditions and a same content of $OrgNO_3$ cannot be expected a priori. This makes comparison between experiments difficult.

c)      ON hydrolysis depends on the character of the ON and only tertiary ON are supposed to hydrolyse fast (Hu et al. 2011). The fraction of tertiary ON depends on the formation process. E.g. oxidation of β-pinene with $NO_3$ led to only a small fraction of tert. ON, so hydrolysis is not so important in case of VOC oxidation by $NO_3$ (Boyd et al. 2015), whereas photochemical production can lead to large fractions of tert. ON and strong hydrolysis (Browne et al. 2013, Takeuchi and Ng, 2019).

Taking a) - c) into account we come to different conclusions as Referee 3. In contrast to the opinion of referee #3 we assess our results as being well within the range of findings reported in the literature (e.g.  Day et al., 2010; Liu et al., 2012; Browne et al., 2013; Jacobs et al., 2014; Rindelaub et al., 2016, 2015; Boyd et al., 2015; Bean and Hildebrandt Ruiz, 2016, Takeuchi and Ng, 2019). All found fast and substantial hydrolysis for photochemically formed particulate ON.

2)      We agree with referee #3 that our descriptions of the determination of effective uptake coefficients (Section 2.4) needed some more details on the experiments. This is described in our response to the specific comment 3.b and we accordingly modified section 2.4.

3)      We went through the manuscript and removed grammatical errors and tried to improve the language.

Specific comments

1. Page 5 line 12. It would be useful to include the amount of ozone added and the steady state ozone concentration in Table 1.

We modified and extended Table 1.

2. Page 6 line 16. Were the seed particles dried before being injected into the chamber? I would assume not but it is not clear from the manuscript. Please specify.

The particles were dried and this is now described in the experimental section (p.6, line 26, in the revised manuscript)

3. Page 8 section 2.4
a. Line 25. What does "finite vapor pressure" mean?

We reformulated that sentence and skipped the notation finite pressure (p.9, line 17 and line 20, in the revised manuscript).

 b. I do not fully understand how these experiments were conducted. From this section, it appears that experiments without seeds (and no organic aerosol formation via nucleation)

were compared with experiments with seeds (ammonium sulfate particles injected) to determine the effective uptake coefficient.

Reply:
No, the uptake experiments were done in one run, starting with the non-seeded chamber. When steady state was reached i.e. stable HOM production, seed particles were added over a few hours to the chamber. Since the chemical production was not affected by adding seed particles, the surface of seed particles provided an increasing sink for the HOM, leading to lower concentrations compared to the non-seeded start. To achieve a sufficient dynamic range of HOM concentrations, a certain level of β-pinene was needed, that caused some NPF in the non-seeded begin of the experiment. The NPF required extrapolation to zero surface in order to calculate $c^0(HOM)$. $c^0(HOM)$ is the concentration in absence of particles which is only determined by (same) production and wall loss. The wall loss coefficients of HOM were determined in independent experiments at lower concentration levels.
We reformulated section 2.4 to better describe how the experiments were conducted. A short paragraph was put in front of Section 2.4 where the experimental procedure is shortly described ==(p. 9, line 1-8, in the revised manuscript)==.

My understanding from the experimental section is that seed particles were only added in "experiment series 3" (i.e., experiments to determine effective uptake coefficient), and no seed particles were added in all other experiments. Presumably, organic aerosol formation via nucleation took place in all other experiments. However, according to Table 1, the b-pinene mixing ratio used in "experiment series 3" was the same as other all experiments. If so, shouldn't nucleation also took place in ""experiment series 3", and that particles (organic particles) would be present in the system even though no ammonium sulfate particles were added (see page 13 line 12)? If this is the case, one needs to consider uptake onto pure organic particles? Please describe and discuss these clearly in the revised manuscript.

Reply:
Yes, the referee is correct, seed particles were added only in experiment series 3. Regarding the effects of NPF in determining $y_{eff}$: uptake of HOM by pure organic particles was certainly considered. The surface concentration ($S_P$) of purely organic particles was $0.2 \times 10^{-3}$ m$^2$/m$^3$ while $S_P$ increased up to $1.2 \times 10^{-3}$ m$^2$/m$^3$ after adding seeds. Plots 1/c(HOM) vs. $S_P$ where linear over the whole range. Or in other words, the pure organic particles matched the behavior of the coated ones. To make this clearer we modified section 2.4 and added a short paragraph in Section 2.4 (p 9, ==line 1-8, p.10, line 24 - 31 and p.11, line 1-2, in the revised manuscript==).
We discussed the procedure of determining $\gamma_{eff}$  also section 3.5 (==p.14, line17-22, in the revised manuscript==)

Particle formation by NPF was desired in experiment series 4 in order to determine OrgNO$_3$ by AMS. The experiments were started at low NO$_X$ to get nucleation and then NO$_X$ was stepwise increased. This seemed easier to us in organic aerosols than in aerosols with seeds. Since HOM form SOA by solely condensation, the organic matrix is the same in SOA formed in NPF and on neutral ammonium sulfate seeds. As stated, the resulting SOA yields shown in Figure 7 were about the same as yields observed in seeded experiments in Sarrafzadeh et al. 2016. In experiment 1 NPF wouldn't affect the results, but in presence of NO$_X$ NPF is strongly suppressed anyhow (Wildt et al., 2014, Sarrafzadeh et al., 2016). So,

NPF is only unwanted in experiment series 2, where one would like to have the HOM totally in the gas-phase in order to determine the expected composition of the SOA by HOM-PP, HOM-ON, HOM-ACC. In the low $NO_X$ cases some NPF took place in series 2, but again with increasing $NO_X$ NPF was suppressed. In cases where NPF took place, loss to particles was corrected as described in supplement section S3.

4. Page 12, line 12. The authors noted that the highly-oxidized C<10 nitrates were observed with increasing NOx and that "supposedly, they did not arise from gas-phase chemistry but were formed at the walls". Please elaborate. What mechanisms at the walls? If there is chemistry on the walls, how would this affect section 2.4 (determination of uptake coefficient) if there is also some sort of wall memory?

As described in manuscript, these compounds had a different composition and showed a different time behavior (increased with time independent on the photochemistry). They cannot affect the uptake coefficients, because the uptake coefficients were determined at $[NOX]_{SS}$ = 4 ppb (β-pinene) or less (α-pinene). As stated in the manuscript and can be seen in Figure S5 (supplement) at these $NO_X$ levels such compounds were unimportant. They could have some effect on the SOA yields at the highest $NO_X$, but that effect is at maximum their contribution to the total HOM. The chamber was flushed for at least on day between experiments until the compounds were below detection limit.
We believe that we gave the phenomenon sufficient attention and showed that it does not much affect our results.
We further weakened the statement that they are formed at the wall (p.14, line 6-8, in the revised manuscript).

5. Page 13, line 16, please also indicate (e.g., in Figure 7) the organic mass concentration, as SOA yield is also highly dependent on organic mass.

We don't understand this comment. The paragraph at page 13 around line 16 deals with uptake coefficients and Figure 5. We are of course aware that SOA yields depend on OA mass, but this not our question here. Experiments in Figure 7 where all performed at the same conditions but changing $NO_X$. The average SOA load was 16±5 µg/m$^3$, ranging from 11 µg/m$^3$ to 23 µg/m$^3$ and this information was added to the caption of Figure 7, in the revised manuscript.

6. Page 14, lines 1-2. Are these mass concentrations in the gas and particle phases consistent with equilibrium partitioning of HNO3?

Reply:
We obviously did not specify sufficiently what we meant. We modified the text (p.15, line 9-11, in the revised manuscript).
Equilibrium partitioning of $HNO_3$ is difficult to predict for SOA at 60%RH. There is some water in the particles and the system will be highly non-ideal. In any case at 24 ug/m$^3$ $HNO_3$ (<8 ppb) the amount in the particulate phase will be orders of magnitude smaller than the gas-phase concentration and an upper limit of 0.1 ug/m$^3$ is well within expectations by thermodynamics. ($HNO_3$ has a vapor pressure at RT of about 5000 Pa.) However, for our considerations it is only important that we determined the OrgNO$_3$ mass correctly and do not falsely count OrgNO3 as inorganic nitrate.

7. Page 16, line 9. There are many more studies. For example, see review and references in Ng et al. (ACP, 2017). Some more recent studies, for example, Claflin and Ziemann (J. Phys. Chem. A, 2018), are also relevant.

We added more references including Ng et al., 2017, as well as Claflin and Ziemann, 2018,

8. Page 16 line 16. A recent study by Takeuchi and Ng (ACP, 2019) also reported on the ON formed by photooxidation of monoterpenes.

We added Takeuchi and Ng, 2019 to the reference list.

9. Page 16 line 23. Please provide citations for this statement "We found contributions between 0 % and 2.7 % by AMS, which is within the range of most other data reported in the literature but at the lower end.".

We modified that passage and added references. (p.17, ==line 5-16, in the revised manuscript==)

10. Page 17 line 9. It was stated that "Our findings are in agreement with observations by Lee et al. (2016b) in a field study." My understanding is that the study by Lee et al. was conducted in a rural environment, presumably with very low level of NOx. The ambient conditions were quite different from laboratory conditions employed in this study. Please justify why an agreement would be expected between results in Lee et al. and this study.

We specified more clearly what we are going to compare (p.18, ==line 25-26, in the revised manuscript==).

11. Page 18-19, section 4.3. The fraction of OrgNO3 is much lower in this study than Zhao et al. and other studies in literature. The authors attributed this to potential hydrolysis of organic nitrates in the particle phase as experiments in Zhao et al. were conducted at much lower RH. However, a recent study by Takeuchi and Ng (ACP,2019), conducted at similar RH to this study, showed that the fraction of organic nitrates in the particles is also much higher than that reported in this study, and the fraction or organic nitrates undergoing hydrolysis was constrained. More analysis should be conducted here to evaluate why the value reported in this study is much lower than prior literature.

Reply
We think there is a misunderstanding. OrgNO$_3$ is not the same as particulate ON. It is the nitrate carried by ON. We clarified that throughout the manuscript. We added a note to section 2.3 (==p.8, line 16-17, in the revised manuscript==), reformulated section 4.1. in large parts ==(p.18 line 2-15, in the revised manuscript)== and modified the first paragraph of section 4.2 (==p.18, line 22-23, in the revised manuscript==).
We basically show that HOM will dominate the composition of particulate phase in our experiment, because of their large $\gamma_{eff}$. Therefore we can predict from the HOM composition, i.e. HOM-PP, HOM-ON and HOM-ACC, the ON and therewith OrgNO$_3$ that should be expected in the particulate phase. We modified the text at several instances showing and discussing now mass concentrations of HOM-ON and all other HOM in new Figure 3 and

new Figure 4. From new Figures 3 and 4 it should now become clear that we got ON mass fractions up to several 10% in the particulate phase. This is in agreement with many other studies, including Takeuchi and Ng, 2019. The discrepancy we discussed was in OrgNO$_3$ expected from HOM and directly measured by AMS, where we find 60-80% loss of the nitrate function. This is somewhat higher than observations of Takeuchi and Ng, 2019, but in agreement with expectations discussed in Boyd et al., 2015, Browne et al. 2013, Fisher et al., 2016.

References

Boyd, C. M., Sanchez, J., Xu, L., Eugene, A. J., Nah, T., Tuet, W. Y., Guzman, M. I., and Ng, N. L.: Secondary organic aerosol formation from the β-pinene+NO$_3$ system: effect of humidity and peroxy radical fate, Atmos. Chem. Phys., 15, 7497-7522, 10.5194/acp-15-7497-2015, 2015.

Browne, E. C., Min, K. E., Wooldridge, P. J., Apel, E., Blake, D. R., Brune, W. H., Cantrell, C. A., Cubison, M. J., Diskin, G. S., Jimenez, J. L., Weinheimer, A. J., Wennberg, P. O., Wisthaler, A., and Cohen, R. C.: Observations of total RONO$_2$ over the boreal forest: NO$_x$ sinks and HNO$_3$ sources, Atmos. Chem. Phys., 13, 4543-4562, 10.5194/acp-13-4543-2013, 2013.

Claflin, M. S., and Ziemann, P. J.: Identification and Quantitation of Aerosol Products of the Reaction of beta-Pinene with NO3 Radicals and Implications for Gas- and Particle-Phase Reaction Mechanisms, Journal of Physical Chemistry A, 122, 3640-3652, 10.1021/acs.jpca.8b00692, 2018.

Fisher, J. A., Jacob, D. J., Travis, K. R., Kim, P. S., Marais, E. A., Chan Miller, C., Yu, K., Zhu, L., Yantosca, R. M., Sulprizio, M. P., Mao, J., Wennberg, P. O., Crounse, J. D., Teng, A. P., Nguyen, T. B., St. Clair, J. M., Cohen, R. C., Romer, P., Nault, B. A., Wooldridge, P. J., Jimenez, J. L., Campuzano-Jost, P., Day, D. A., Hu, W., Shepson, P. B., Xiong, F., Blake, D. R., Goldstein, A. H., Misztal, P. K., Hanisco, T. F., Wolfe, G. M., Ryerson, T. B., Wisthaler, A., and Mikoviny, T.: Organic nitrate chemistry and its implications for nitrogen budgets in an isoprene- and monoterpene-rich atmosphere: constraints from aircraft (SEAC4RS) and ground-based (SOAS) observations in the Southeast US, Atmos. Chem. Phys., 16, 5969-5991, 10.5194/acp-16-5969-2016, 2016

Hu, K. S., Darer, A. I., and Elrod, M. J.: Thermodynamics and kinetics of the hydrolysis of atmospherically relevant organonitrates and organosulfates, Atmos. Chem. and Phys., 11, 8307-8320, 10.5194/acp-11-8307-2011, 2011.

Rindelaub, J. D., McAvey, K. M., and Shepson, P. B.: The photochemical production of organic nitrates from alpha-pinene and loss via acid-dependent particle phase hydrolysis, Atmos. Environ., 100, 193-201, 10.1016/j.atmosenv.2014.11.010, 2015

Takeuchi, M., and Ng, N. L.: Chemical composition and hydrolysis of organic nitrate aerosol formed from hydroxyl and nitrate radical oxidation of α-pinene and β-pinene, Atmos. Chem. Phys., 19, 12749-12766, 10.5194/acp-19-12749-2019, 2019

---

## Author Response (AR2)

**Suggestions for revision or reasons for rejection (will be published if the paper is accepted for final publication)**

The manuscript is much improved over the previous version, particularly section 4, with no conclusions or findings have been changed. The addition to figures 3 and 4 and Table 1 greatly improve the understanding of the work. The responses to all 3 reviewers were thorough and thoughtful and thus I only have 3 very minor comments relating to newly added material.

We thank the reviewer for the helpful comments. We gratefully followed the suggestions and we will acknowledge his efforts in the acknowledgement of our manuscript-

**Minor comments**

Page 18, line 21: can you define "small loads of SOA" with a value? Or do you mean atmospheric concentrations of SOA?

Answer: we performed the experiment at organic load < 10 ug/m$^3$ and we added this information into the manuscript.

Page 19, line 7: could also consider citing Compernolle et al., ACP 2011 (https://www.atmos-chem-phys.net/11/9431/2011/), another commonly used vapor pressure estimation model, but not strictly necessary.

Answer:reference to model EVAPORATION was added in section 4.2

Figure 3, caption: too much discussion is occurring here. Please limit to description. I.e.: probably move everything beyond the first two sentences into the discussion section, except this sentence: "Dashed and dotted lines save to guide…". This will also help unbury this sentence which is currently easy to miss although it's very important for understanding the figure.

Answer:We removed all interpretations form the Figure caption as suggested and integrated the removed into §1 of section 3.3.

Figure 4, caption: are the lines here also to guide the eye or are they a fit?

Answer: They serve to guide the eye. We added this info to the caption.

**The grammar was much improved, although minor issues remain.**

We again thank reviewer 2 for all the efforts to help us to improve the manuscript.

Page 1, line 27: "0.5 in average" should be "0.5 on average"

done

Page 1, line 30: "character the final" should be "character of the final"

done

Page 7, line 23: should "uncertainty on our results" be "uncertainty to our results"?

done

Page 7, line 27: there shouldn't be a comma here: "section S2 shows,"

done

Page 9, line 21: "of less" should be "of less than". And the numbers in "m2 m-3" should be superscript

both done

page 10, line 19-20: "particle free" and "particle containing" should be "particle-free" and "particle-containing"
done

page 11, line 3: shouldn't be a comma here: "is only valid, if"
done

page 11, line 5: "likely due to fact" should be "likely due to the fact"
done

page 12, line 23: "increase of peaks with odd molecular masses odd peaks", I think the "odd peaks" part shouldn't be there?
done

Page 12, line 31: "as function of" should be "as a function of"
done

Page 15, line 32: "enumerator" should be "numerator"
done

Page 16, line 25: should "estimate an possible effect" be "estimate the possible effect"
done

Page 17, line 15: the "mass' " of "cmass' " should be super-scripted
done, thoughout the manuscript

Page 18, line 4: "with same efficiency" should be "with the same efficiency"
done

Page 19, line 15: "despite of the" should be "despite the"
done

Page 20, line 5: "in presence of" should be "in the presence of"
done

Page 20, line 22: "despite of" should be "despite"
done

Page 21, line 19: "HOMM-ACC" should be "HOM-ACC"
done

Table 1, footnote b: "ratio in steady state" should be "ratio at steady state"
done

Figure 2 caption: use of "µg/m-3" is incorrect: can't have the divide symbol and the -3. Suggest re-writing as "µg m-3" to be consistent with usage in rest of caption, if using the divide should not have the minus symbol.
done

References

[revised manuscript text omitted]